# Embracing Unknown Step by Step:
# Towards Reliable Sparse Training in Real World

**Bowen Lei**                                    *bowenlei@stat.tamu.edu*
*Department of Statistics*
*Texas A&M University*

**Dongkuan Xu**                                  *dxu27@ncsu.edu*
*Department of Computer Science*
*North Carolina State University*

**Ruqi Zhang**                                   *ruqiz@purdue.edu*
*Department of Computer Science*
*Purdue University*

**Bani Mallick**                                 *bmallick@stat.tamu.edu*
*Department of Statistics*
*Texas A&M University*

**Reviewed on OpenReview:** *https://openreview.net/forum?id=Db5c3Wxj9E*

## Abstract

Sparse training has emerged as a promising method for resource-efficient deep neural networks (DNNs) in real-world applications. However, the reliability of sparse models remains a crucial concern, particularly in detecting unknown out-of-distribution (OOD) data. This study addresses the knowledge gap by investigating the reliability of sparse training from an OOD perspective and reveals that sparse training exacerbates OOD unreliability. The lack of unknown information and the sparse constraints hinder the effective exploration of weight space and accurate differentiation between known and unknown knowledge. To tackle these challenges, we propose a new unknown-aware sparse training method, which incorporates a loss modification, auto-tuning strategy, and a voting scheme to guide weight space exploration and mitigate confusion between known and unknown information without incurring significant additional costs or requiring access to additional OOD data. Theoretical insights demonstrate how our method reduces model confidence when faced with OOD samples. Empirical experiments across multiple datasets, model architectures, and sparsity levels validate the effectiveness of our method, with improvements of up to **8.4%** in AUROC while maintaining comparable or higher accuracy and calibration. This research enhances the understanding and readiness of sparse DNNs for deployment in resource-limited applications. Our code is available on: `https://github.com/StevenBoys/MOON`.

## 1 Introduction

Sparse training is one popular method for achieving resource efficiency in deep neural networks (DNNs), and it is receiving growing attention when considering the resource-limited real-world applications of DNNs (Bellec et al., 2017; Evci et al., 2020; Yuan et al., 2021). By maintaining sparse weights throughout the training process, sparse training accelerates the training of DNNs, saves training memory, and produces sparse models with dense performance levels (Mocanu et al., 2018; Dettmers & Zettlemoyer, 2019; Evci et al., 2020; Liu et al., 2022a), which has been applied to an increasing number of tasks (Yuan et al., 2021; Sokar et al., 2021; Bibikar et al., 2022). There remains, however, a critical question that needs to be answered before sparse

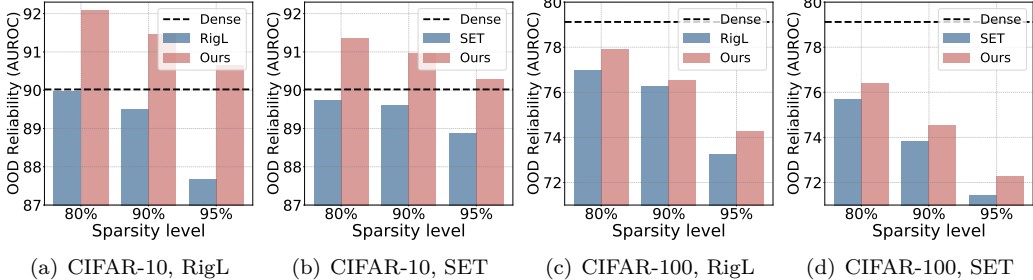

(a) CIFAR-10, RigL    (b) CIFAR-10, SET    (c) CIFAR-100, RigL    (d) CIFAR-100, SET

Figure 1: OOD reliability (measured by AUROC (%), the higher the better) of the ResNet-18 produced by dense and sparse training (RigL & SET) on CIFAR-10/100. Compared to dense training (black line), sparse training (blue bar) has a smaller AUROC, indicating sparse training exacerbates the unreliability on OOD data. Our MOON (red bar) improves AUROC and OOD detection.

training can be safely introduced in real-world applications: how reliable the sparse model is in the real world?

Reliable models in real-world scenarios possess the ability to recognize situations in which they are likely to be wrong, such as identifying unknown OOD data, and refrain from providing incorrect responses (Wang et al., 2021b; Vaze et al., 2021; Du et al., 2022). However, the investigation of OOD detection within the context of sparse training remains unexplored, creating an important knowledge gap that hinders a comprehensive assessment of the readiness of DNNs for deployment. Models with weak OOD detection are prone to feign understanding and provide arbitrary guesses on unrecognizable OOD data, leading to safety concerns (Liu et al., 2020; Hsu et al., 2020; Du et al., 2022; Wang et al., 2021b) in scenarios, e.g., automated healthcare and self-driving cars (Rasheed et al., 2022; De Silva & Mycroft, 2022).

In this work, we investigate for the *first* time the reliability of sparse training from the OOD perspective, and find that sparse training exacerbates the OOD unreliability. Figure 1 compares AUROC (i.e., a metric for OOD detection, the higher the better) (Yang et al., 2022) between dense and sparse training, e.g., RigL (Evci et al., 2020) and SET (Mocanu et al., 2018). The AUROC is smaller for sparse training (blue bar) compared to dense training (black line), indicating that sparse training weakens the OOD detection capacity. In order to address the OOD unreliability of sparse training, we raise the following questions:

**Question 1.** *Why sparse training exacerbates the OOD unreliability in DNNs?*

The underlying determinant of weak OOD detection is the lack of unknown information, and sparse constraints hinder the effective communication of unknown concepts with DNNs. In order to recognize OOD samples, the model needs to distinguish between the known and the unknown. However, it has been shown that DNNs tend to focus on the known due to the close nature of the training process and ignore the rest (Bendale & Boult, 2016; Padhy et al., 2020; Wang et al., 2021b). There are studies trying to encourage DNNs to learn more about the unknown. But they usually focus on dense training. When moving to sparse training, the sparsity cuts off update routes and produces spurious local minima (Evci et al., 2019; Sun & Li, 2021; He et al., 2022), which makes the exploration of the weight space more challenging, and thus more difficult to search for and find reliable models (Lei et al., 2023b). Thus, the sparse training exacerbates the OOD unreliability in DNNs.

**Question 2.** *How can we leverage unknown information under sparse constraints to effectively guide weight space exploration to OOD-reliable state?*

To improve OOD reliability, it is crucial to utilize unknown information to guide model training. The sparsity constraint increases the difficulty of exploring the weight space and prevents the unknown information from guiding the model training to an OOD-reliable state. Efforts have been made to exploit unknown information in dense DNNs (Thulasidasan et al., 2019; Wang et al., 2021b; Roy et al., 2022). However, DAC (Thulasidasan et al., 2019) is limited to the treatment to label noise and lacks a comprehensive guide to OOD detection. EOW-Softmax (Wang et al., 2021b) only indirectly and inadequately exploits the unknown and incurs considerable additional costs. Moreover, HOD (Roy et al., 2022) require access to OOD data, which is often a non-trivial task. To address these issues and adapt to the challenges in sparse training, we propose a new outlier-exposure-free training loss that extracts unknown information from hard ID samples and then directly tells the DNNs this unknown information, which gives an effective guide for weight space exploration under sparse constraints with only a slight increase in computational and storage burden.

***Question 3.*** *How can we address the confusion between known and unknown information in sparse training to avoid providing misleading guidance for weight space exploration?*

Despite the possibility of communicating unknown information, sparse DNNs are more likely to confuse known and unknown information, resulting in poor performance and reliability. DNNs show performance degradation in sparse training with high sparsity (Mocanu et al., 2018; Evci et al., 2020; Lei et al., 2023b), especially in the early training stage, which indicates the challenge of understanding knowable content under sparsity constraints. Therefore, to avoid confusion and provide proper guidance for training, we also design an auto-tuning strategy that allows the DNN to mainly discover the known in the early stage and embrace the unknown in the later stage step by step. In addition, we propose a voting scheme that combines information from multiple models for a more comprehensive understanding of the known and unknown.

In summary, our contributions are summarized as follows:

- We for the first time investigate how sparse training influences OOD reliability and analyze the hidden reasons. We find that sparse training exacerbates the OOD unreliability of DNNs.

- We propose a new model-agnostic unknown-aware sparse training method (`MOON`) to improve real-world reliability, which allows the model to be aware of what it does not know via a loss modification, auto-tuning strategy, and a voting scheme at little additional cost.

- We provide theoretical insights on how our `MOON` method reduces the confidence of the model when faced with OOD samples and improves OOD reliability.

- Empirically, we conduct extensive experiments on multiple benchmark datasets, model architectures, and sparsity. Our `MOON` improves the AUROC by up to **8.4%** and successfully upholds the accuracy, calibration, and other pertinent metrics associated with sparse training, thereby complementing existing research and further enhancing the efficacy and applicability of the sparse training paradigm.

## 2 Related Work

**Sparse Training.** As models continue to grow in size, there is an increasing focus on sparse training, where sparse weights are maintained during training to accommodate resource-limited real-world deployments (Mocanu et al., 2018; Bellec et al., 2018; Huang et al., 2023). To find a sparse model with good performance, the sparse topology is updated by pruning and growth steps after every fixed number of iterations (Evci et al., 2020; Huang et al., 2022). A variety of sparse training methods have been investigated and various pruning and growth criteria, such as weights and gradient magnitude, have been developed (Mostafa & Wang, 2019; Dettmers & Zettlemoyer, 2019; Evci et al., 2020; Jayakumar et al., 2020; Liu et al., 2021b; Özdenizci & Legenstein, 2021; Zhou et al., 2021; Schwarz et al., 2021; Yin et al., 2022; Lei et al., 2023a). Existing work mainly targets saving more resources and at the same time maintaining accuracy. However, despite saving resources, sparse training can result in more difficult training due to the cut-off update routes and the generation of spurious local minima, raising safety concerns on both ID and OOD data.

**Confidence Calibration.** DNNs' reliability on ID data is increasingly important, which usually refers to whether the confidence of DNNs on ID data is well calibrated (Guo et al., 2017; Nixon et al., 2019; Zhang et al., 2020; Wu et al., 2023). Prior work has shown that DNNs tend to be over-confident (Guo et al., 2017; Rahaman et al., 2021; Patel et al., 2022; Zhu et al., 2023), misleading humans and raising safety concerns. Research has been conducted to improve confidence calibration, where widely-used methods include temperature scaling (Guo et al., 2017), Mixup (Zhang et al., 2017), label smoothing (Szegedy et al., 2016), and Bayesian methods (Gal & Ghahramani, 2016; Ashukha et al., 2020). Recently, ID reliability in sparse training has also been studied in Sup-tickets (Yin et al., 2022) and CigL (Lei et al., 2023b).

**OOD Detection & Open-set Classification.** For real-world tasks, reliability on OOD data, i.e., OOD detection capability, is also critical due to the presence of unknown categories. It is also referred to as open-set classification when ID classification is required as well (Yang et al., 2021b; Vaze et al., 2021). When these unknown categories are given, a reliable model should detect them and refuse to answer, rather than giving random guesses. Research has been devoted to OOD detection, including post-processing methods (Hendrycks

& Gimpel, 2016; Liang et al., 2017; Liu et al., 2020), training time regularization (DeVries & Taylor, 2018; Hsu et al., 2020; Du et al., 2022), and training with outlier exposure (Hendrycks et al., 2018; Yu & Aizawa, 2019; Yang et al., 2021a). However, existing OOD reliability studies have mainly focused on dense training and have not yet fully explored sparse training settings.

**OOD reliability of sparse DNNs.** Despite the lack of exploration of sparse training, there are studies on the OOD reliability of sparse DNNs. The effect of random pruning on the sparse DNNs with fixed sparsity patterns is studied (Liu et al., 2022b), while sparse training has dynamic sparsity patterns that provide better performance compared to fixed patterns. It is also investigated how to obtain deep ensembles that outperform dense models through sparse training, which requires a non-trivial additional cost (Liu et al., 2021a). Thus, generating a single sparse network through sparse training remains irreplaceable when resources are limited. In addition, efforts have been devoted to OOD reliability of pruning methods, which start with a dense model and gradually prune the weights, reducing the difficulty of exploring the weight space (Ayle et al., 2022; Cheng et al., 2023). In contrast, sparse training maintains a high level of sparsity throughout the training process, which cuts off a large portion of the route and produces more spurious local minima, thus making weight space exploration more challenging and having different properties from pruning methods (Chen et al., 2022b).

**Extra Dimension.** Adding an extra dimension to the output of the K-way classification model has been studied and used to relax closed-world setting to open-world setting (Hsu et al., 2020; Mozannar & Sontag, 2020; Wang et al., 2021b; Verma & Nalisnick, 2022; Liu et al., 2023). Specifically, the close-world setting assumes the test data has the same distribution as the training data, which is not always true in the real world. The open-world setting, on the other hand, allows for unknown classes in the test data. Previous work has used extra dimension to abstain from making decisions on noisy labels (Thulasidasan et al., 2019) or to improve OOD detection in dense DNNs (Wang et al., 2021b; Roy et al., 2022). However, they do not directly exploit the unknown information and incur extra costs (Wang et al., 2021b), or require exposure to OOD data (Roy et al., 2022). In addition, they all ignore how to add extra dimensions in sparse training.

## 3 Method

We propose a new sparse training method `MOON` to improve the real-word reliability of DNNs, i.e., to provide effective detection for OOD data under sparsity constraints with comparable or higher ID accuracy and reliability. Importantly, `MOON` can be seamlessly integrated as a plug-in method and combined with existing OOD detection methods. Specifically, we add an extra dimension in the output probability and propose a new unknown-aware loss with an auto-tuning strategy to gradually encourage DNNs to consider more unknown information. Then, we combine information from multiple models into the output model via a voting scheme so that the output model has a comprehensive view of what is known and what is unknown, providing reliable predictions.

### 3.1 Tell Model What It Doesn't Know

In general, given the data $\{x_i, y_i\}_{i=1}^N$ where $y \in \{1, \cdots, K\}$, the model is forced to classify new input data into K predefined classes by the widely used cross-entropy loss, even if the model is uncertain about the class of the samples or has never seen this class before (Wang et al., 2021b). We argue that ignorance of unknown information during weight space exploration, i.e., the sample for which the model gives incorrect predictions ($\widehat{y}_i \neq y_i$), is the cause of this unreliability. Sparse constraints further make the exploration more challenging and thus require better guidance from unknown information. To solve these problems, we utilize K+1-way formulation and propose a new unknown-aware loss to tell the model what is unknown to it during sparse training. Unlike existing methods, we directly tell DNNs the unknown, providing an effective guide to cope with the challenges posed by sparsity.

**K+1-way formulation:** To store the unknown information, inspired by previous work on dense DNNs with extra dimensions (Thulasidasan et al., 2019; Wang et al., 2021b; Roy et al., 2022), we add an extra dimension in the softmax formulation, allowing the model to provide a $K + 1$-dimensional output probability $f(x_i) = (p_1, \cdots, p_K, p_{K+1})$ for each $x_i$, unlike the previous $K$-way formulation. The first $K$ dimensions

$(p_1, \cdots, p_K)$ represent the model's beliefs about class attribution, while the $K + 1$-th dimension $p_{K+1}$ is designed to store unknown information. Ideally, the model gives larger probabilities for the true class dimension $p_{y_i}$ when the model knows the input samples very well. Otherwise, the model provides larger $p_{K+1}$ and smaller $(p_1, \cdots, p_K)$ to avoid unreliable random guesses in the $K$ classes.

**Unknown-aware Loss Modification:** To incorporate unknown information into the model, we design an unknown-aware loss shown in Eq. (1). On the one hand, when the model knows the sample $x_i$ well and makes correct predictions (i.e., $\widehat{y}_i = y_i$), we choose the broadly-used cross-entropy loss for $x_i$ to encourage larger $p_{y_i}$. On the other hand, if the model does not have enough understanding of harder sample $x_i$ and makes wrong predictions (i.e., $\widehat{y}_i \neq y_i$), we use a new-designed loss to increase reliability. Specifically, in the new loss, we add a larger weight $(1 + \frac{w}{1 + w p_{K+1}})$ to encourage the model to focus more on the samples it predicts incorrectly. To reduce the loss, the model can determine the correct class (i.e., give a larger $p_{y_i}$) or provide a larger $p_{K+1}$ such that $(p_1, \cdots, p_K)$ becomes smaller, indicating that the model is uncertain about its prediction.

$$L(x_i) = \begin{cases} -\log p_{y_i} & \widehat{y}_i = y_i \\ -(1 + \frac{w}{1 + w p_{K+1}}) \log p_{y_i} & \widehat{y}_i \neq y_i \end{cases} \tag{1}$$

## 3.2 Do not Frustrate Model in Early Stage

We find that telling the model too much about the unknowns at the beginning can lead to more severe unreliability (see Section 5.7), especially in cases where the weight space is insufficiently explored (e.g., highly sparse models). It is as if too much emphasis on unknowns frustrates the model and prevents it from finding out what it can know. Thus, inspired by (Thulasidasan et al., 2019), we design an auto-tuning strategy for $w$ in the unknown-aware loss (i.e., Eq. (1)) to step-by-step inform the model about the unknown information, which is outlined in Algorithm 1. In this way, $p_{y_i}$ and $p_{K+1}$ can better help the model learn and store both the known and unknown information.

**In the early stage** (i.e., epoch $t <= T_e$), we fix $w$ to zero so that the new loss is equivalent to the cross-entropy loss and no unknown information is emphasized. DNNs usually learn easy samples in the early stage of training and tend to fit the hard ones later (Wang et al., 2021a). Thus, in the beginning, the model does not learn the data well and usually makes many wrong predictions. With insufficient exploration of the weight space, a large $w$ will unnecessarily put more emphasis on harder samples and prevent the model from learning easy samples, leading to poor reliability. The model is like a new learner, and it is advisable not to point out its mistakes too much, in case it gets frustrated.

**After the early stage** (i.e., epoch $t > T_e$), we start telling the model what is unknown. We keep $w$ as a constant at each epoch and increase it from the initial value $w_i$ to the final value $w_f$ after each epoch via a linear scheduler. **The initial $w_i$ is based on** the $\beta = (1 - p_{K+1})L(x_i)$ and calculated in the early stage. On the one hand, if we get smaller $p_{K+1}$ and larger $L(x_i)$, we can know that the model hasn't learned the data well while it is over-confident about

---

**Algorithm 1:** $w$ Auto-tuning Scheduler

**Input:** Total epoch $T$, current epoch $t$, unknown-free epoch $T_e$, probability $\{p_i\}_{i=1}^{K+1}$, unknown-free loss $L$, init factor $r$, final weight $w_f$, smoothing factor $\alpha$.

**Output:** The weight $w$ for the unknown-aware loss.

Initialization:

$w = 0$, $\widetilde{\beta} = 0$, and $\delta = -1$

**if** $t < T_e$ **then**

    Prepare information for the initial weight:
    $\beta = (1 - p_{K+1})L(x_i)$
    Use moving average $\widetilde{\beta} = (1 - \alpha)\widetilde{\beta} + \alpha\beta$

**else if** $\delta = -1$ **then**

    Calculate the initial weight $w_i = \frac{\widetilde{\beta}}{r}$ and $w = w_i$
    Calculate the step size $\delta = \frac{w_f - w_i}{T - T_e}$ of the scheduler

**else**

    Calculate weight at epoch $t$:
    $w = w_i + (t - T_e) \cdot \delta$

**end**

---

its wrong predictions. Thus, we will set a larger $\beta$ and $w_i$, in this case, to emphasize more on the unknown for reliability. On the other hand, if we get larger $p_{K+1}$ and smaller $L(x_i)$, we can know that the model learns the data well and is cautious about its output. Therefore, we can encourage the model to find out more about what it can know with smaller $\beta$ and $w_i$. For the initial factor $r$, it is used to control the initial value

$w_i$ to avoid $w$ being too large. **For the final** $w_f$, it is a pre-determined hyper-parameter, which is usually larger than $w_i$. At each epoch after $T_e$, we set a linear increase scheduler $w = w_i + (t - T_e) \cdot \delta$ where the step size $\delta = (w_f - w_i)/(T - T_e)$, allowing more attention on the unknown at the later stage of the training.

### 3.3 Vote for Known and Unknown: Weight Averaging

We further design a voting strategy that combines information from multiple models to give the output model a more comprehensive sense of what it knows and what it does not know. Specifically, at the end of the proposed unknown-aware training (e.g., after 80% training epochs), we collect models at each epoch and use the weight averaging method (Izmailov et al., 2018; Wortsman et al., 2022) to uniformly average them into one output model. In this way, we let multiple models decide what is known and unknown together, allowing a fuller understanding of the learned knowledge and more reliable predictions.

## 4 Theoretical Insight

We provide a couple of theoretical insights to demonstrate that our model-agnostic unknown-aware sparse training method `MOON` can produce more reliable predictions in the face of out-of-distribution (OOD) data. The unknown information is extracted from hard in-distribution (ID) data, which explains the outlier-exposure-free property of `MOON`. Detailed proofs are presented in the Appendix. Without loss of generality, suppose the goal is a two-way classification based on $\{x_i, y_i\}_{i=1}^N$ where $y_i \in \{1, 2\}$. We have a deep neural network $f = g \circ h$, which can be seen as a composition of feature mapping function $h$ and the softmax classification function $g$.

We first show how we can extract the unknown from the hard ID data. The following analysis is based on Assumption 4.1, which has been widely used in several studies of OOD detection (Lee et al., 2018; Ahuja et al., 2019; Morteza & Li, 2022).

**Assumption 4.1.** (Gaussian Mixture Feature Space): The feature mapping function $h$ maps the input ID data to a Gaussian mixture $v_1 \mathcal{N}(\mu_1, \Sigma_1) + v_2 \mathcal{N}(\mu_2, \Sigma_2)$. Specifically, when $y = 1$, we have $h(x) \sim \mathcal{N}(\mu_1, \Sigma_1)$. And when $y = 2$, we have $h(x) \sim \mathcal{N}(\mu_2, \Sigma_2)$.

We define the definition of unreliability for a set of ID data.

**Definition 4.2.** (Unreliability) For samples whose features are around $h(x_0)$, i.e., $D(\epsilon_0) = \{(x, y); ||h(x) - h(x_0)|| < \epsilon_0\}$, the predictions are unreliable around $x_0$ if for any $0 < \epsilon < \epsilon_0$, we can find $\eta > 0$ such that

$$\mathbf{E}_{D(\epsilon)}[\max_{c \in \{1,2\}} \mathcal{N}(h(x); \mu_c, \Sigma_c)] - \mathbf{E}_{D(\epsilon)}[1\{\hat{y} = y\}] > \eta. \tag{2}$$

*Remark* 4.3. Definition 4.2 gives a mathematical expression for the ID unreliability in feature space (i.e., the discrepancy between confidence and accuracy) around $h(x_0)$. The first term $\mathbf{E}_{D(\epsilon)}[\max_{c \in \{1,2\}} \mathcal{N}(x; \mu_c, \Sigma_c)]$ denotes the expected confidence around $h(x_0)$, and the second term $\mathbf{E}_{D(\epsilon)}[1\{\hat{y} = y\}]$ refers to the expected accuracy. For a threshold (e.g., $\eta$), we consider predictions to be unreliable if the discrepancy between confidence and accuracy cannot be reduced below $\eta$ no matter how we alter $\epsilon$. This extends the definition of unreliability from the global aspect to the local region around $h(x_0)$ in feature space and helps to describe unreliability within hard samples.

**Insight 4.4.** (Unreliability) Suppose we have $(x_1, y_1)$ from class 1 and $D_2 = \{(x, y); ||h(x) - h(x_1)|| < \epsilon, y = 2\}$ from class 2. Then, unreliability can occur around $h(x_1)$.

*Remark* 4.5. The setting in Insight 4.4 is common for the hard ID samples, resulting in unreliability. Samples with large feature differences within two classes are easy to classify, with both high accuracy and confidence. While samples with minor differences can puzzle the model, with low accuracy and high confidence, resulting in over-confidence.

**Insight 4.6.** (Hard-ID Reliability) Suppose we have the same $(x_1, y_1)$ and $D_2$ as in Insight 4.4. If the model is trained with our `MOON` method and the extra dimension successfully stores the unknown information, the unreliability can be solved, i.e., for any $\eta > 0$, we can find $0 < \epsilon < \epsilon_0$ such that

$$\mathbf{E}_{D(\epsilon)}[\max_{c \in \{1,2\}} \mathcal{N}(h(x); \mu_c, \Sigma_c)] - \mathbf{E}_{D(\epsilon)}[1\{\hat{y} = y\}] < \eta. \tag{3}$$

*Remark* 4.7. Insight 4.6 shows that our `MOON` method reduces the confidence level and decreases the discrepancy, solving the unreliability issue when the model is faced with some hard ID samples.

We further show how OOD detection can benefit from the unknown information from hard ID data.

**Insight 4.8.** (OOD Reliability) Suppose we achieve Hard-ID Reliability in Insight 4.6 with our `MOON` method, we can have lower confidence on OOD data, implying stronger OOD detection.

*Remark* 4.9. Insignt 4.8 show that OOD detection can be enhanced by the unknown information extracted from the hard ID data. Existing work has found that ID data contain information for the OOD samples (Du et al., 2022). It is found that ID samples from the low-likelihood region of the class-conditional distribution, i.e., hard-to-detect ID data, are similar to OOD samples in the feature space, which is similar to the scenario described in Insight 4.4. Although we do not use OOD data in the training, these hard-to-detect ID samples can be viewed as pseudo-OOD data. With our `MOON` method, the model knows which hard ID samples it does not know and provides lower confidence for these samples (i.e., pseudo-OOD data). Since the real OOD data are neighbors of the pseudo-OOD data in the feature space, the model also gives a low confidence for the real OOD data.

## 5 Experiments

We perform a comprehensive empirical evaluation of our `MOON` , where we add our `MOON` to the training of multiple benchmark datasets, model architectures, and sparsities, and compare it with the original sparse training results.

**Datasets & Model Architectures:** For in-distribution (ID) data, we include four benchmark datasets: MNIST (Deng, 2012), CIFAR-10 and CIFAR-100 (Krizhevsky et al., 2009) and ImageNet-2012 (Russakovsky et al., 2015). For out-of-distribution (OOD) data, we follow the setups of OpenOOD (Yang et al., 2022) (more details in Appendix B.1). For model architectures, we choose ResNet-18 to train MNIST, CIFAR-10, and CIFAR-100, and choose ResNet-50 to train ImageNet-2012 (He et al., 2016). We repeat all experiments 3 times and calculate the mean and standard deviation.

**Training Settings:** For sparse training, we study two widely-used methods, e.g., RigL (Evci et al., 2020) and SET (Mocanu et al., 2018), and a variety of sparsities, including 80%, 90%, 95%, and 99%, which sufficiently reduces the cost and is of great interest in real-world deployments. We also examine the performance of dense training to show the broad applicability of our method (see Appendix A.3).

**Implementations:** We follow the settings of OpenOOD (Yang et al., 2022). We choose SGD with momentum and use the cosine annealing learning rate scheduler. We train all the models for 100 epochs. For the batch size, we set it to 128 in MNIST, CIFAR-10, and CIFAR-100, and 32 in ImageNet.

**Comparison Metrics:** To measure OOD detection capability, we chose AUROC and FPR-95 (Yang et al., 2022). For AUROC, it calculates the area under the receiver operating characteristic (ROC) curve, where a larger AUROC suggests better OOD detection. For FPR-95, it measures the false positive rate (FPR) at a true positive rate (TPR) of 95%, where a lower FPR-95 implies better OOD detection. To compare ID reliability, we choose ECE (Guo et al., 2017) and test accuracy. ECE measures the discrepancy between confidence and true accuracy, with lower ECE indicating higher ID reliability.

### 5.1 OOD Detection in Sparse Training

In this section, we show that our `MOON` can improve OOD detection in sparse training, including 80%, 90%, 95%, and 99% sparsities using RigL. Since our `MOON` requires no additional OOD data and training, to fairly compare the OOD reliability, we incorporate our `MOON` into the widely-used post-process OOD detection methods MSP (Hendrycks & Gimpel, 2016) and compare the results MSP.

**For CIFAR-10 based models**, we examine its OOD detection capacity on two near OOD data, i.e., CIFAR-100 (Chandola et al., 2009) and TIN (Han et al., 2022)), and four far OOD data, i.e., MNIST (Deng, 2012), SVHN (Netzer et al., 2011), Texture (Ahmed & Courville, 2020), Places365 (Guo et al., 2017). As shown in Table 1, the columns named "NearOOD" and "FarOOD" represent the average detection scores

Table 1: Comparison of OOD detection by AUROC (%) (↑) between MOON +MSP (*M-MSP*) and MSP for CIFAR-10 in sparse training. Our MOON leads to larger AUROC on each OOD data of CIFAR-10, showing its ability to improve reliability on OOD data.

| | | CIFAR-100 | TIN | NearOOD | MNIST | SVHN | Texture | Places365 | FarOOD |
|---|---|---|---|---|---|---|---|---|---|
| 80% | MSP | 87.43 (0.2) | 88.92 (0.2) | 88.17 (0.2) | 92.92 (0.8) | 91.81 (0.7) | 89.35 (0.3) | 89.44 (0.2) | 90.88 (0.5) |
| | *M-MSP* | **89.93** (0.1) | **91.14** (0.1) | **90.53** (0.1) | **95.00** (0.7) | **93.82** (0.5) | **91.80** (0.2) | **90.82** (0.3) | **92.86** (0.4) |
| 90% | MSP | 87.78 (0.2) | 89.12 (0.1) | 88.45 (0.1) | 91.82 (0.9) | 91.48 (0.5) | 87.90 (0.3) | 88.93 (0.2) | 90.03 (0.4) |
| | *M-MSP* | **88.92** (0.2) | **90.34** (0.2) | **89.63** (0.2) | **95.39** (0.8) | **93.09** (0.4) | **91.09** (0.4) | **89.92** (0.2) | **92.37** (0.4) |
| 95% | MSP | 86.04 (0.1) | 87.30 (0.1) | 86.67 (0.1) | 93.74 (0.6) | 86.09 (0.8) | 85.26 (0.3) | 87.68 (0.2) | 88.19 (0.5) |
| | *M-MSP* | **88.04** (0.1) | **89.39** (0.2) | **88.71** (0.1) | **94.76** (0.7) | **93.35** (0.8) | **89.86** (0.3) | **88.51** (0.3) | **91.62** (0.5) |
| 99% | MSP | 80.94 (0.2) | 82.75 (0.2) | 81.84 (0.2) | 88.34 (0.8) | 85.75 (0.7) | 83.43 (0.2) | 80.45 (0.2) | 84.49 (0.5) |
| | *M-MSP* | **83.75** (0.1) | **85.11** (0.1) | **84.43** (0.1) | **91.96** (0.8) | **89.01** (0.8) | **86.44** (0.3) | **83.38** (0.2) | **87.70** (0.5) |

of the near OOD data and the far OOD data, respectively. As shown in Table 1, the AUROC value drops with increasing sparsity, which implies the unreliability issues in sparse training. Our MOON obtains a larger AUROC on each OOD data of CIFAR-10, where the improvement of AUROC can be up to 8.4%. This indicates MOON 's ability to improve the reliability on OOD data in sparse training.

**For the model trained on ImageNet-2012**, we examine its OOD detection ability on four near OOD data, i.e., Species (SP) (Torralba et al., 2008), iNaturalist (IN) (Shorten & Khoshgoftaar, 2019), OpenImage-O (OO) (Li et al., 2021), ImageNet-O (IO) (Sun et al., 2022), and two far OOD data, i.e., Texture (TX) (Ahmed & Courville, 2020), MNIST (MN) (Deng, 2012). As shown in Figure 2, the red and blue bars represent our MOON and MSP, respectively. We can see that our MOON provides larger AUROC and smaller FPR-95 almost on all of the OOD data of ImageNet-2012, showing its effective OOD detection in sparse training.

**To show the general applicability of our method**, we also incorporate our MOON into other broadly-used post-processing OOD detection methods, e.g., ODIN (Liang et al., 2017), EBO (Liu et al., 2020), KNN (Sun et al., 2022), and KLM (Hendrycks et al., 2022), and compare the results with those using only ODIN and EBO. We take sparse models on CIFAR-10 using RigL as an example. First, we consider 99% sparsity while considering ODIN and EBO as baselines. As shown in Table 2, our MOON leads to larger AUROC and smaller FPR-95 on each OOD data compared to ODIN and EBO. Then, for KNN and KLM, we also experiment on different sparsity levels in-

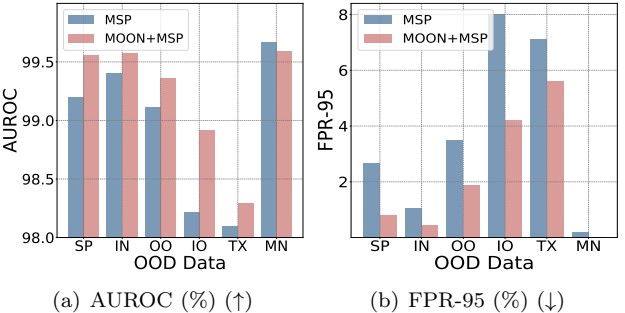

(a) AUROC (%) (↑)     (b) FPR-95 (%) (↓)

Figure 2: Comparison of OOD detection by AUROC (%) (↑) and FPR-95 (%) (↓) between MOON +MSP and MSP for ImageNet-2012 using RigL (90%). MOON leads to larger AUROC and smaller FPR-95 on OOD data of ImageNet-2012, showing improved OOD reliability in sparse training.

cluding 80%, 90%, 95%, and 99%. As shown in Table 3, our MOON leads to a larger AUROC on each OOD data compared to KNN and KLM. This shows that the ability of our MOON to improve OOD reliability is consistent under sparsity constraints for different OOD detection methods.

## 5.2 Comparison with Calibration Method

Prior work shows that calibration methods can help with OOD detection in dense DNNs (Wang et al., 2021b; Yang et al., 2022). To show the ineffectiveness of existing calibration methods for OOD detection in sparse training, we also compare our MOON to calibration methods including temperature scaling (Lee et al., 2018), Mixup (Zhang et al., 2017), and CigL (Lei et al., 2023b) using RigL (90%) for OOD detection capability. For our MOON , we report the results of "MOON + ODIN". As shown in Figure 3, the red and blue hexagons

Table 2: Comparison of OOD detection by AUROC (%) (↑) & FPR-95 (%) (↓) between MOON +ODIN (*M-ODIN*), MOON +EBO (*M-EBO*) and baseline post process methods (i.e., ODIN and EBO) for CIFAR-10 using RigL (99% sparsity). Our MOON leads to larger AUROC and smaller FPR-95 on each OOD data, showing its ability to improve reliability on OOD data in sparse training.

|  |  | CIFAR-100 | TIN | NearOOD | MNIST | SVHN | Texture | Places365 | FarOOD |
|---|---|---|---|---|---|---|---|---|---|
| AUROC | ODIN | 84.09 (0.2) | 86.98 (0.3) | 85.54 (0.2) | 96.82 (0.5) | 89.11 (0.7) | 87.47 (0.3) | 85.49 (0.2) | 89.72 (0.4) |
| | *M-ODIN* | **90.59** (0.3) | **92.64** (0.2) | **91.61** (0.2) | **99.21** (0.7) | **93.91** (0.8) | **93.26** (0.2) | **93.35** (0.2) | **94.93** (0.5) |
| | EBO | 83.73 (0.3) | 86.93 (0.3) | 85.33 (0.3) | 94.68 (0.6) | 85.71 (0.5) | 85.09 (0.2) | 85.44 (0.2) | 87.73 (0.4) |
| | *M-EBO* | **85.64** (0.2) | **88.55** (0.3) | **87.09** (0.2) | **97.10** (0.7) | **87.35** (0.4) | **87.11** (0.3) | **88.55** (0.3) | **90.03** (0.4) |
| FPR-95 | ODIN | 67.99 (0.8) | 61.54 (0.5) | 64.76 (0.6) | 19.40 (1.6) | 65.32 (1.7) | 56.58 (0.6) | 62.56 (0.7) | 50.97 (1.1) |
| | *M-ODIN* | **61.23** (0.8) | **56.00** (0.6) | **58.62** (0.7) | **11.58** (1.8) | **57.67** (2.1) | **51.81** (0.5) | **54.09** (0.7) | **43.79** (1.3) |
| | EBO | 69.42 (0.7) | 61.15 (0.6) | 65.29 (0.6) | 34.49 (1.4) | 78.09 (1.2) | 67.64 (0.6) | 61.23 (0.7) | 60.36 (1.0) |
| | *M-EBO* | **63.30** (0.7) | **55.85** (0.5) | **59.58** (0.6) | **16.37** (1.6) | **77.70** (0.7) | **60.83** (0.7) | **51.48** (0.8) | **51.60** (0.9) |

Table 3: Comparison of OOD detection by AUROC (%) (↑) between MOON +KNN, MOON +KLM and baseline post-process methods (i.e., KNN and KLM) for CIFAR-10 using RigL (80%, 90%, 95%, 99% sparsity). Our MOON leads to larger AUROC on each OOD data, showing its ability to improve reliability on OOD data in sparse training.

|  |  | CIFAR-100 | TIN | NearOOD | MNIST | SVHN | Texture | Places365 | FarOOD |
|---|---|---|---|---|---|---|---|---|---|
| 80% | KNN | 89.50 | 91.12 | 90.31 | 94.11 | 93.76 | 92.69 | 90.83 | 92.85 |
| | MOON +KNN | **90.16** | **92.11** | **91.14** | **95.58** | **93.77** | **94.02** | **92.47** | **93.96** |
| | KLM | 77.29 | 78.98 | 78.14 | 83.65 | 86.22 | 80.29 | 77.89 | 82.01 |
| | MOON +KLM | **78.92** | **81.92** | **80.42** | **89.29** | **86.44** | **84.78** | **80.23** | **85.18** |
| 90% | KNN | 88.49 | 90.54 | 89.51 | 92.93 | 92.12 | 92.06 | 89.98 | 91.77 |
| | MOON +KNN | **89.40** | **91.32** | **90.36** | **95.04** | **93.97** | **92.99** | **91.73** | **93.43** |
| | KLM | 77.60 | 79.66 | 78.63 | 87.9 | 85.58 | 80.93 | 76.64 | 82.76 |
| | MOON +KLM | **79.21** | **81.29** | **80.25** | **89.38** | **89.94** | **83.33** | **80.04** | **85.67** |
| 95% | KNN | 88.10 | 89.87 | 88.98 | 94.85 | 93.10 | 92.60 | 89.50 | 92.51 |
| | MOON +KNN | **88.36** | **90.34** | **89.35** | **94.98** | **93.62** | **92.91** | **90.51** | **93.01** |
| | KLM | 76.56 | 78.34 | 77.45 | 87.84 | 78.86 | 77.67 | 76.15 | 80.13 |
| | MOON +KLM | **79.06** | **81.26** | **80.16** | **89.64** | **84.04** | **83.59** | **79.35** | **84.16** |
| 99% | KNN | 81.87 | 84.18 | 83.03 | 92.69 | 86.00 | 86.15 | 82.96 | 86.95 |
| | MOON +KNN | **82.97** | **84.78** | **83.88** | **96.62** | **95.64** | **89.15** | **83.03** | **91.11** |
| | KLM | 72.27 | 74.47 | 73.37 | 78.44 | 65.96 | 74.22 | 73.64 | 73.06 |
| | MOON +KLM | **75.40** | **76.63** | **76.01** | **85.03** | **82.32** | **78.51** | **74.90** | **80.19** |

represent the FPR-95 of MOON and calibration methods, respectively. We can see that the red hexagons are smaller than the blue hexagons, indicating better OOD detection from MOON compared to calibration methods.

### 5.3 ID Test Accuracy and Reliability in Sparse Training

In this section, we show that, in addition to improving OOD detection, our MOON can also maintain or improve the accuracy and reliability on ID data. We incorporate our MOON into sparse training (RigL) and compare it with the original sparse training results without MOON.

**For ID Test Accuracy**, as shown in Figure 4 (a)-(b), the red and blue curves represent our MOON +RigL and RigL, respectively. The red curve is usually higher or equal to the blue curve, implying that our MOON can maintain comparable or higher test accuracy on the ID data.

**For ID Reliability**, we choose expected calibration error (ECE) (Guo et al., 2017), a widely-used measure of the discrepancy between a model's confidence and true accuracy, where a lower ECE indicates better confidence calibration and higher reliability. As shown in Figure 4 (c)-(d), the red and blue curves represent our MOON+RigL and RigL, respectively, where the colored ares represent the 95% confidence intervals. The

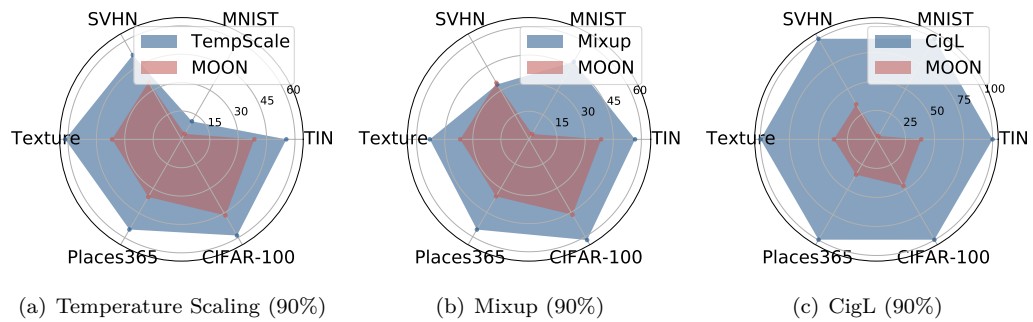

(a) Temperature Scaling (90%)      (b) Mixup (90%)      (c) CigL (90%)

Figure 3: Comparison of OOD detection by FPR-95 (%) (↓) on CIFAR-10 between `MOON` and other calibration methods using RigL (90%). The red hexagons (`MOON` ) are smaller than the blue hexagons (other calibration methods), indicating a better OOD detection using `MOON` compared to (a) Temperature Scaling, (b) Mixup, and (c) CigL.

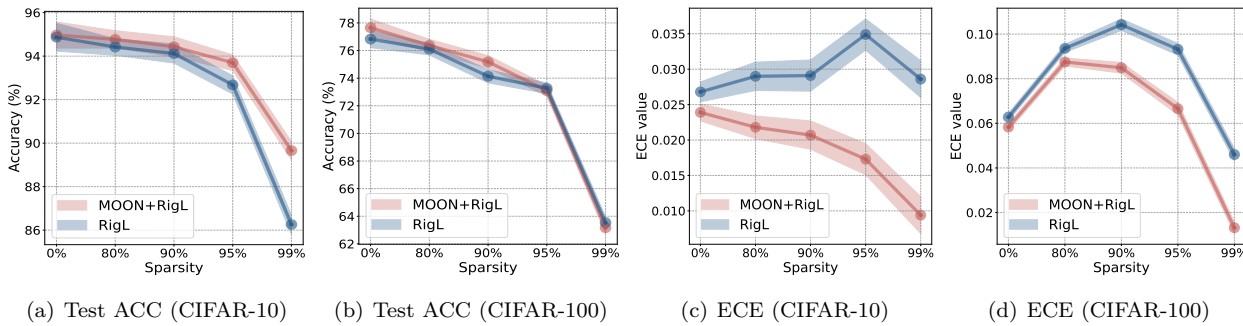

(a) Test ACC (CIFAR-10)    (b) Test ACC (CIFAR-100)    (c) ECE (CIFAR-10)    (d) ECE (CIFAR-100)

Figure 4: Comparison of performance on ID data by ECE (↓) and test accuracy (ACC) (%) (↑) between `MOON` and RigL. Our `MOON` leads to smaller ECE and maintains comparable or higher test accuracy on ID data, showing its ability to improve reliability on ID data.

red curve is usually lower than the blue curve. This shows that our `MOON` reduces the ECE, implying an improvement in the ID reliability of our `MOON`.

Table 4: Comparison of reliability of ID and OOD data between `MOON` +MSP (*M-MSP*) and MSP for CIFAR-10 in sparse training using SET. Our `MOON` leads to smaller ECE and comparable or higher accuracy (ACC) (%) on ID data, and smaller FPR-95 (%) on each OOD data compared to baseline method MSP, showing its ability to improve reliability on ID and OOD data.

| | | ID Data | | OOD Data (FPR-95 (%) (↓)) | | | | | |
|---|---|---|---|---|---|---|---|---|---|
| | | ECE(↓) | ACC (↑) | CIFAR-100 | TIN | MNIST | SVHN | Texture | Places365 |
| 80% | MSP | 0.031 (0.002) | 94.03 (0.2) | 64.01 (0.5) | 60.64 (0.4) | **46.62** (1.2) | 66.43 (1.4) | 64.50 (0.7) | 60.63 (0.4) |
| | *M-MSP* | **0.020** (0.001) | **94.80** (0.2) | **62.72** (0.4) | **58.63** (0.3) | 47.63 (1.5) | **60.68** (1.2) | **59.10** (0.8) | **59.39** (0.3) |
| 90% | MSP | 0.032 (0.001) | 93.94 (0.2) | 64.70 (0.3) | 62.52 (0.4) | 50.45 (1.6) | 67.50 (1.7) | 64.13 (0.6) | 63.33 (0.4) |
| | *M-MSP* | **0.019** (0.001) | **94.44** (0.3) | **63.33** (0.3) | **60.15** (0.5) | **43.02** (1.2) | **58.81** (2.2) | **61.26** (0.6) | **60.79** (0.4) |
| 95% | MSP | 0.035 (0.002) | 92.56 (0.3) | 67.62 (0.5) | 65.18 (0.4) | 51.79 (1.7) | 67.95 (1.5) | 67.46 (0.7) | 65.44 (0.5) |
| | *M-MSP* | **0.020** (0.003) | **93.59** (0.3) | **63.69** (0.5) | **60.20** (0.5) | **41.82** (1.9) | **62.02** (1.1) | **57.29** (0.8) | **60.85** (0.4) |
| 99% | MSP | 0.025 (0.001) | 87.58 (0.2) | 74.98 (0.4) | 73.27 (0.5) | 54.12 (1.5) | 74.44 (1.2) | 74.36 (0.7) | 74.03 (0.3) |
| | *M-MSP* | **0.008** (0.002) | **88.67** (0.2) | **71.88** (0.5) | **69.59** (0.5) | **45.14** (1.4) | **71.10** (1.1) | **65.73** (0.9) | **71.61** (0.3) |

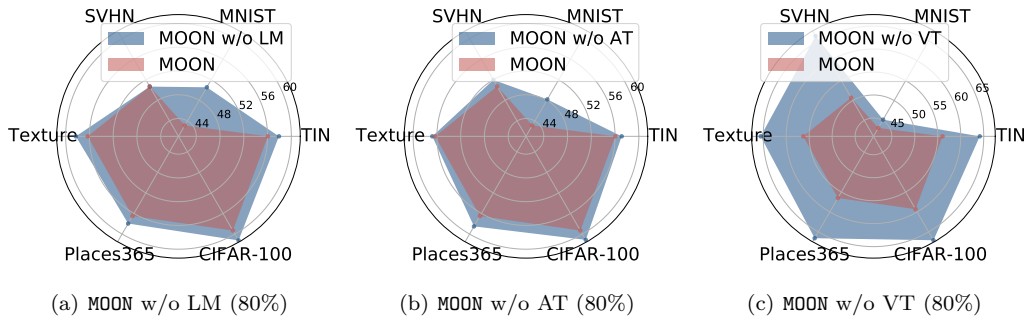

Figure 5: Ablation studies: comparison of FPR-95 (%) (↓) between MOON , MOON w/o LM, MOON w/o AT, and MOON w/o VT on OOD data of CIFAR-10. MOON produces lower FPR-95 compared to the other three methods.

Table 5: Comparison of OOD detection by AUROC (%) (↑) between MOON +MSP (*M-MSP*), LogitNorm+MSP (L-MSP), and VOS+MSP (V-MSP) for CIFAR-10 in sparse training. Our MOON leads to larger AUROC on each OOD data of CIFAR-10, showing its ability to improve OOD reliability.

|  |  | CIFAR-100 | TIN | NearOOD | MNIST | SVHN | Texture | Places365 | FarOOD |
|---|---|---|---|---|---|---|---|---|---|
| | L-MSP | 87.71 | 89.60 | 88.66 | 94.74 | 93.29 | 89.36 | 90.79 | 92.05 |
| 80% | V-MSP | 87.57 | 89.04 | 88.31 | 91.82 | 90.93 | 89.42 | 88.60 | 90.19 |
| | *M-MSP* | **89.93** | **91.14** | **90.53** | **95.00** | **93.82** | **91.80** | **90.82** | **92.86** |
| | L-MSP | 87.81 | 89.41 | 88.61 | 95.37 | 91.26 | 90.24 | 89.85 | 91.68 |
| 90% | V-MSP | 87.48 | 88.58 | 88.03 | 92.97 | 92.17 | 88.93 | 88.81 | 90.53 |
| | *M-MSP* | **88.92** | **90.34** | **89.63** | **95.39** | **93.09** | **91.09** | **89.92** | **92.37** |
| | L-MSP | 87.35 | 88.58 | 87.96 | 93.36 | 86.36 | 87.79 | 88.49 | 89.03 |
| 95% | V-MSP | 87.12 | 87.98 | 87.55 | 93.64 | 86.40 | 88.10 | 87.89 | 89.01 |
| | *M-MSP* | **88.04** | **89.39** | **88.71** | **94.76** | **93.35** | **89.86** | **88.51** | **91.62** |
| | L-MSP | 82.60 | 84.52 | 83.56 | 91.75 | 87.56 | 84.16 | 82.68 | 86.54 |
| 99% | V-MSP | 82.22 | 83.09 | 82.65 | 84.75 | 89.99 | 82.75 | 82.26 | 84.69 |
| | *M-MSP* | **83.75** | **85.11** | **84.43** | **91.96** | **89.01** | **86.44** | **83.38** | **87.70** |

## 5.4 Comparison with Other Sparse Training Method

Apart from the comparison based on RigL, we also include sparse training results using SET (Mocanu et al., 2018) on CIFAR-10 at 80%, 90%, 95%, 99% sparsities. As shown in Table 4, on OOD data, our MOON also leads to smaller FPR-95, indicating improved OOD reliability from our MOON. On ID data, our MOON brings smaller ECE and comparable or higher accuracy (ACC), implying comparable or improved ID reliability and performance. This shows our MOON is effective in different sparse training methods.

## 5.5 Comparison with Other Training-time Regularization Method

Since our loss modification can be viewed as a new type of training-time regularization, we also add additional experiments on other dense training-time regularization methods including LogitNorm (Wei et al., 2022a) and VOS (Du et al., 2022). As shown in Table 5, our method MOON has larger AUROC compared to LogitNorm and VOS, indicating better OOD reliability from our method in sparse training.

## 5.6 Comparison with Other Extra-dimension Method

Since our loss modification also belongs to the extra-dimension method, we further compare our MOON with another extra-dimension method, i.e., DAC (Thulasidasan et al., 2019) regarding the reliability of ID and OOD data. As shown in Table 6, our MOON also provides larger AUROC when given OOD data. When faced with ID data, our MOON leads to smaller ECE and comparable or higher accuracy (ACC). This suggests that

Table 6: Comparison of reliability of ID and OOD data between MOON and DAC for CIFAR-10 and CIFAR-100. Our MOON leads to smaller ECE and comparable or higher accuracy (ACC) on ID data, and larger AUROC on each OOD data, showing its ability to improve reliability on ID and OOD data.

| Data | Method | ID Data | | OOD Data (AUROC ($\uparrow$)) | | | | | |
|---|---|---|---|---|---|---|---|---|---|
| | | ECE($\downarrow$) | ACC($\uparrow$) | CIFAR-100/10 | TIN | MNIST | SVHN | Texture | Places365 |
| CIFAR-10 | DAC | 0.0302 | 94.43 | 86.17 | 87.96 | 89.92 | 87.26 | 89.32 | 88.48 |
| | MOON | **0.0239** | **94.97** | **89.93** | **91.14** | **95.00** | **93.82** | **91.80** | **90.82** |
| CIFAR-100 | DAC | 0.0873 | 76.03 | 76.99 | 80.54 | 74.43 | 77.70 | **77.89** | 77.86 |
| | MOON | **0.0583** | **77.65** | **78.82** | **82.38** | **80.28** | **81.57** | 77.62 | **79.65** |

Table 7: Ablation studies: comparison of ECE ($\downarrow$) and accuracy (ACC, $\uparrow$) between MOON , MOON w/o LM, MOON w/o AT, and MOON w/o VT on CIFAR-100. MOON produces lower ECE values and comparable or higher accuracy.

| Method | 80% | | 90% | | 95% | | 99% | |
|---|---|---|---|---|---|---|---|---|
| | ECE ($\downarrow$) | ACC (%) ($\uparrow$) | ECE ($\downarrow$) | ACC (%) ($\uparrow$) | ECE ($\downarrow$) | ACC (%) ($\uparrow$) | ECE ($\downarrow$) | ACC (%) ($\uparrow$) |
| MOON w/o LM | 0.0272 | 94.68 | 0.0282 | 94.40 | 0.0307 | 93.44 | 0.0207 | 89.63 |
| MOON w/o AT | 0.0250 | **94.81** | 0.0258 | 94.51 | 0.0308 | 93.26 | 0.0200 | 89.50 |
| MOON w/o VT | 0.0296 | 92.97 | 0.0256 | 91.81 | 0.0330 | 90.23 | 0.0096 | 85.38 |
| MOON | **0.0218** | 94.77 | **0.0207** | **94.53** | **0.0173** | **93.70** | **0.0094** | **89.66** |

our MOON makes better use of the additional dimension to encourage the model to think about the unknown, thus improving real-world reliability while maintaining ID performance and reliability.

## 5.7 Ablation Studies

We do ablation studies to demonstrate the importance of each component in our MOON , where we train CIFAR-100 using our MOON without the unknown-aware loss modification (MOON w/o LM), MOON without the auto-tuning strategy (MOON w/o AT), and MOON without the average-based voting scheme (MOON w/o VT), respectively. The OOD detection is shown by FPR-95 for each OOD data of CIFAR-100 in Figure 5, and the ID results are summarized in Table 7.

**MOON w/o LM**: As shown in Figure 5 (a), the blue hexagon (MOON w/o LM) is larger than the red hexagon (MOON), indicating a decrease in OOD reliability without the unknown-aware loss modification. Table 7 also shows an increased ECE and decreased ID reliability in MOON w/o LM.

**MOON w/o AT**: Similarly, as depicted in Figure 5 (b), the blue hexagon (MOON w/o AT) is larger than the red hexagon (MOON), indicating a decrease in OOD reliability in the absence of the auto-tuning strategy. Table 7 also shows an increased ECE and decreased ID reliability in MOON w/o AT.

**MOON w/o VT**: As summarized in Figure 5 (c), the blue hexagon (MOON w/o VT) is larger than the red hexagon (MOON), indicating a decrease in OOD reliability when no average-based voting scheme is present. Table 7 further shows a decreased accuracy and increased ECE in MOON w/o VT.

## 6 Conclusion and Limitations

We investigate for the first time the reliability of sparse training from an OOD perspective, which jointly considers OOD reliability and efficiency and has important implications for real-world DNN applications. We observe that sparse training causes more severe unreliability, with useful unknown information being ignored in the training. Thus, we propose a new unknown-aware sparse training method, MOON, to enhance OOD detection under sparsity constraints, with little additional cost and no extra OOD data requirement.

We provide theoretical insights to demonstrate the lower confidence brought by `MOON` on OOD samples. We also conduct extensive experiments on multiple benchmark datasets, architectures, and sparsities, showing that our `MOON` can improve AUROC by up to 8.4%. Our `MOON` can be used as a drop-in replacement for existing sparse training methods, which fills an important gap in sparse training and improves the effectiveness and applicability of the sparse training paradigm.

Despite the enhanced OOD reliability achieved in sparse training, `MOON` does not address this matter comprehensively from a broader reliability perspective, including robust generalization (Silva & Najafirad, 2020; Özdenizci & Legenstein, 2021; Chen et al., 2022a) and adaptation (Emam et al., 2021; Alayrac et al., 2022; Hu et al., 2022), which are equally vital aspects of reliability and deserve further investigation (Tran et al., 2022). Additionally, the effectiveness of `MOON` remains unexplored in the context of widely employed large generative models (Ouyang et al., 2022; Xu et al., 2023; Croitoru et al., 2023). On the one hand, as model size expands, novel changes in model characteristics may arise (Wei et al., 2022b). On the other hand, given the prevalence of non-classification tasks within LLMs, the exploration of how to evaluate their reliability remains an uncharted domain.

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

# A  Appendix: Additional Experimental Results

## A.1  More Sparse Training Results

To further demonstrate the effectiveness of our MOON , apart from the comparison of AUROC (↑) on CIFAR-10 in the main manuscript, we also include more comparison of FPR-95 (↓) between MOON+MSP and MSP for CIFAR-10 in sparse training using RigL. As shown in Table 8, our MOON usually leads to smaller FPR-95, indicating more effective OOD detection from our MOON compared to baseline methods.

Table 8: Comparison of OOD detection by FPR-95 (↓) between MOON+MSP and MSP for CIFAR-10 in sparse training using RigL. Our MOON leads to larger AUROC on each OOD data of CIFAR-10, showing its ability to improve reliability on OOD data.

| | | CIFAR-100 | TIN | NearOOD | MNIST | SVHN | Texture | Places365 | FarOOD |
|---|---|---|---|---|---|---|---|---|---|
| 80% | MSP | 62.97 | 59.71 | 61.34 | 48.16 | 53.70 | 60.35 | 58.59 | 55.20 |
| | MOON+MSP | **59.31** | **56.04** | **57.68** | **39.12** | **50.71** | **56.28** | **56.49** | **50.65** |
| 90% | MSP | 63.71 | 61.54 | 62.62 | 54.85 | 58.33 | 66.76 | 61.43 | 60.34 |
| | MOON+MSP | **62.70** | **58.18** | **60.44** | **33.67** | **54.65** | **57.32** | **58.41** | **51.01** |
| 95% | MSP | 70.38 | 66.96 | 68.67 | 44.19 | 74.57 | 73.01 | 65.37 | 64.28 |
| | MOON+MSP | **65.29** | **61.11** | **63.20** | **39.26** | **52.72** | **63.44** | **63.38** | **54.70** |
| 99% | MSP | 77.34 | 73.71 | 75.53 | 60.45 | 77.77 | 71.54 | 76.49 | 71.56 |
| | MOON+MSP | **70.38** | **67.05** | **68.72** | **44.34** | **72.08** | **62.75** | **68.54** | **61.93** |

To show that the improvement from our MOON is consistent across different sparse training methods, apart from the comparison based on RigL, we also include more sparse training results using SET (Mocanu et al., 2018). As shown in Tables 9, our MOON usually leads to larger AUROC, indicating more effective OOD detection from our MOON compared to baseline methods.

Table 9: Comparison of OOD detection by AUROC (↑) between MOON+MSP and MSP for CIFAR-10 in sparse training using SET. Our MOON leads to larger AUROC on each OOD data of CIFAR-10, showing its ability to improve reliability on OOD data.

| | | CIFAR-100 | TIN | NearOOD | MNIST | SVHN | Texture | Places365 | FarOOD |
|---|---|---|---|---|---|---|---|---|---|
| 80% | MSP | 87.36 | 88.81 | 88.09 | 93.41 | 88.70 | 88.51 | 88.77 | 89.85 |
| | MOON+MSP | **89.52** | **90.81** | **90.16** | **93.64** | **92.47** | **91.25** | **90.46** | **91.96** |
| 90% | MSP | 87.93 | 89.06 | 88.49 | 93.54 | 90.26 | 88.40 | 88.57 | 90.19 |
| | MOON+MSP | **88.85** | **90.06** | **89.45** | **94.29** | **92.34** | **90.70** | **89.61** | **91.74** |
| 95% | MSP | 86.81 | 87.83 | 87.32 | **94.95** | 89.13 | 87.07 | 87.62 | 89.69 |
| | MOON+MSP | **88.02** | **89.33** | **88.68** | 93.78 | **91.96** | **90.09** | **88.56** | **91.10** |
| 99% | MSP | 81.79 | 82.41 | 82.10 | 90.13 | 87.08 | 81.31 | 81.77 | 85.07 |
| | MOON+MSP | **83.10** | **83.84** | **83.47** | **91.13** | **89.14** | **84.92** | **81.90** | **86.77** |

Prior work shows that calibration methods can help with OOD detection in dense DNNs (Wang et al., 2021b; Yang et al., 2022). To show the ineffectiveness of existing calibration methods for OOD detection in sparse training, we also compare our MOON to other confidence calibration methods including temperature scaling (Lee et al., 2018), Mixup (Zhang et al., 2017), and CigL (Lei et al., 2023b) using RigL for OOD detection capability in 99% sparsity. As shown in Figure 6, the red and blue hexagons represent the FPR-95 of MOON and other calibration methods, respectively. We can see that the red hexagons are smaller than the blue hexagons, indicating better OOD detection from MOON compared to other calibration methods.

To check the effectiveness of our MOON at low sparsity level, we add additional experiments at 50%, 60%, and 70% sparsity on CIFAR-10 and CIFAR-100. The detailed results are summarized in Figure 7. Our MOON can consistently improve the OOD reliability at different sparsity levels.

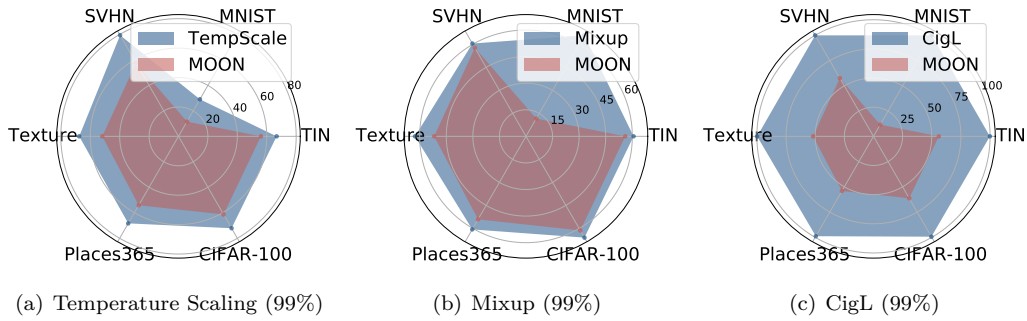

(a) Temperature Scaling (99%)          (b) Mixup (99%)          (c) CigL (99%)

Figure 6: Comparison of OOD detection by FPR-95 (↓) on CIFAR-10 between MOON and other calibration methods using RigL (90%). The red hexagons (MOON) are smaller than the blue hexagons (other calibration methods), indicating a better OOD detection using MOON compared to (a) Temperature Scaling, (b) Mixup, and (c) CigL.

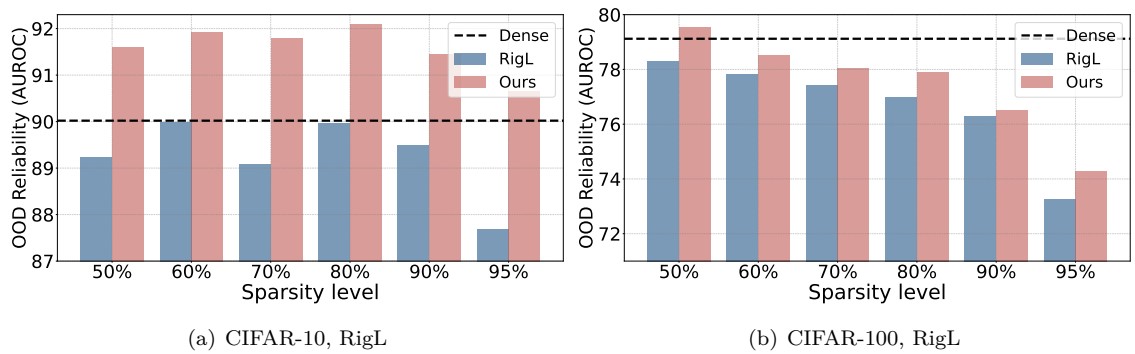

(a) CIFAR-10, RigL          (b) CIFAR-100, RigL

Figure 7: OOD reliability (measured by AUROC (%), the higher the better) of the ResNet-18 produced by dense and sparse training (RigL & SET) on CIFAR-10/100. Compared to dense training (black line), sparse training (blue bar) has a smaller AUROC, indicating sparse training exacerbates the unreliability on OOD data. Our MOON (red bar) improves AUROC and OOD detection.

Table 10: Comparison of OOD detection by FPR-95 (FPR, ↓) & AUROC (AUC, ↑) between MOON and MSP for MNIST. Our MOON leads to smaller FPR-95 and larger AUROC on each OOD data, showing its ability to improve reliability on OOD data.

| | | NotMNIST | FashionMNIST | NearOOD | Texture | CIFAR10 | TIN | Places365 | FarOOD |
|---|---|---|---|---|---|---|---|---|---|
| FPR | MSP | 40.81 | 30.46 | 35.64 | 7.43 | 7.11 | 6.11 | 9.28 | 7.48 |
| | MOON+MSP | **3.57** | **7.08** | **5.32** | **1.76** | **0.91** | **1.41** | **1.32** | **1.35** |
| AUC | MSP | 89.20 | 93.69 | 91.45 | 98.55 | 98.55 | 98.75 | 98.20 | 98.51 |
| | MOON+MSP | **98.07** | **96.54** | **97.31** | **98.86** | **99.18** | **99.06** | **99.12** | **99.05** |

## A.2   More Results on MNIST

To show that the improvement from our MOON is consistent across different benchmark datasets, apart from the comparison on CIFAR-10, CIFAR-100, and ImageNet-2012, we also include more results on MNIST. As shown in Tables 10, our MOON usually leads to larger AUROC and smaller FPR-95, indicating more effective OOD detection from our MOON compared to baseline methods.

Table 11: Comparison of OOD detection by FPR-95 (↓) & AUROC (↑) between `MOON` and baseline post process methods for CIFAR-10. Our `MOON` leads to smaller FPR-95 and larger AUROC on each OOD data, showing its ability to improve reliability on OOD data.

|  |  | CIFAR-100 | TIN | NearOOD | MNIST | SVHN | Texture | Places365 | FarOOD |
|---|---|---|---|---|---|---|---|---|---|
| FPR-95 | MSP | 62.01 | 60.69 | 61.35 | 58.59 | 51.87 | 59.89 | 57.64 | 57.00 |
|  | MOON+MSP | **61.01** | **56.87** | **58.94** | **40.91** | **48.25** | **51.31** | **57.56** | **49.50** |
|  | ODIN | 59.09 | 59.06 | 59.07 | 36.23 | 67.92 | 51.10 | 50.51 | 51.44 |
|  | MOON+ODIN | **48.47** | **42.24** | **45.36** | **8.10** | **33.30** | **32.58** | **40.64** | **28.65** |
|  | EBO | 51.46 | 45.02 | 48.24 | 44.50 | 44.94 | 48.32 | 41.88 | 44.91 |
|  | MOON+EBO | **47.41** | **39.29** | **43.35** | **14.98** | **21.34** | **36.75** | **38.51** | **27.90** |
| AUROC | MSP | 87.11 | 86.62 | 86.87 | 89.91 | 90.88 | 88.72 | 89.03 | 89.64 |
|  | MOON+MSP | **89.06** | **90.61** | **89.83** | **94.11** | **93.83** | **92.30** | **89.97** | **92.55** |
|  | ODIN | 77.68 | 77.33 | 77.51 | 90.91 | 73.32 | 80.70 | 82.55 | 81.87 |
|  | MOON+ODIN | **87.91** | **90.26** | **89.08** | **98.34** | **93.37** | **92.97** | **90.17** | **93.71** |
|  | EBO | 86.15 | 88.58 | 87.36 | 90.59 | 88.39 | 86.85 | 89.60 | 88.86 |
|  | MOON+EBO | **90.05** | **92.36** | **91.20** | **97.30** | **96.15** | **93.34** | **92.04** | **94.71** |

### A.3 OOD Detection in Dense Training

In this section, to show the broad applicability of our `MOON`, we show our `MOON` can improve OOD detection in dense training. The results of baseline methods are based on the scores reported in (Yang et al., 2022).

**For the model trained on CIFAR-10**, we examine its OOD detection ability on two near OOD data (i.e., CIFAR-100, TIN) and four far OOD data (i.e., MNIST, SVHN, Texture, Places365). As shown in Table 11, the columns named "NearOOD" and "FarOOD" represent the average detection scores of the near OOD data and the far OOD data, respectively. We can see that our `MOON` provides smaller FPR-95 and larger AUROC for different near and far OOD data compared to MSP, ODIN, and EBO, where the reduction of FPR-95 can be up to 77.6% and the improvement of AUROC can be up to 27.3%. This demonstrates that our `MOON` can perform OOD detection more effectively compared to the original methods.

**For the model trained on ImageNet-2012**, we examine its OOD detection ability on four near OOD data (i.e., Species, iNaturalist, OpenImage-O, ImageNet-O) and two far OOD data (i.e., Texture, MNIST). More details about the OOD data are in Appendix B.1. As shown in Figure 8, the red and blue bars represent our `MOON` and MSP, respectively. We can see that our `MOON` provides larger AUROC and AUPR, and smaller FPR-95 almost on all of the OOD data of ImageNet-2012, showing its effective OOD detection.

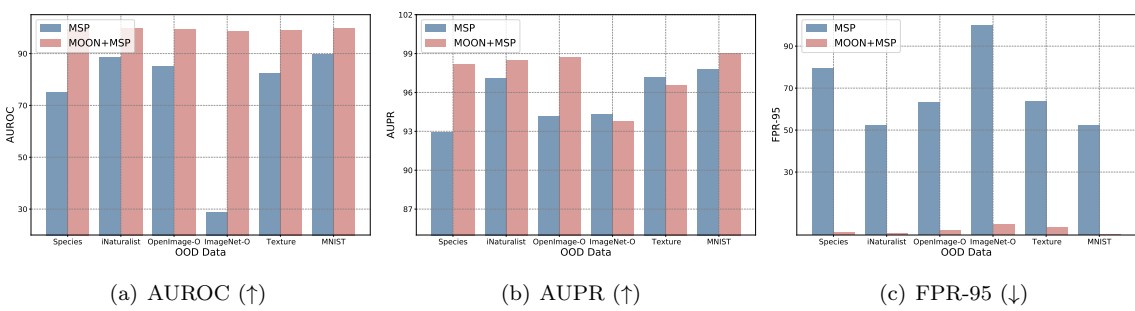

(a) AUROC (↑)   (b) AUPR (↑)   (c) FPR-95 (↓)

Figure 8: Comparison of OOD detection by AUROC (↑), AUPR (↑), and FPR-95 (↓) between `MOON+MSP` and MSP for ImageNet-2012. Our `MOON` leads to larger AUROC and AUPR, and smaller FPR-95 on each OOD data of ImageNet-2012, showing its ability to improve reliability on OOD data.

Table 12: More Ablation Studies for $w_f$ by AUROC ($\uparrow$) in `MOON` with 95% sparse ResNet-18 for CIFAR-10.

| | | CIFAR-100 | TIN | NearOOD | MNIST | SVHN | Texture | Places365 | FarOOD |
|---|---|---|---|---|---|---|---|---|---|
| AUROC | $w_f = 0.1$ | 83.57 | 85.25 | 84.41 | 86.49 | 86.47 | 85.35 | 84.73 | 85.76 |
| | $w_f = 0.5$ | 86.25 | 87.07 | 87.66 | 93.46 | 91.02 | 87.80 | 85.11 | 89.35 |
| | $w_f = 1.0$ | **88.04** | **89.39** | **88.71** | **94.70** | **93.35** | **89.86** | **88.51** | **91.62** |
| | $w_f = 2.0$ | 86.45 | 87.77 | 87.11 | 89.96 | 91.44 | 89.24 | 86.05 | 89.17 |
| | $w_f = 4.0$ | 84.78 | 85.84 | 85.31 | 94.69 | 91.44 | 86.66 | 84.65 | 89.36 |

## A.4 More Ablation Studies for $w_f$

We add additional ablation studies for $w_f$ to test the sensitivity of $w_f$. As summarized in Appendix G, we set $w_f = 1$ for CIFAR-10. We add experiments of 95% sparse ResNet-18 with $w_f = 0.1, 0.5, 2, 4$. The AUROC results are summarized in the table below. We also include them in Appendix A.4. As shown in the Table 12, when we increase $w_f$ from 0.1 to 4, OOD reliability first increases and then decreases after 1. While OOD reliability may be degraded when $w_f$ is too small or too large, OOD reliability is not degraded much when $w_f$ is between 0.5 and 2. The proper $w_f$ value can be found with a simple hyperparameter tuning.

## A.5 Additional Incremental Ablation Study

We add an incremental ablation study in which each component of the method is incremented to the previous component to better intuitively understand the utility of each component. We train 95% sparse ResNet-18 on CIFAR-10 with MOON + RigL. We find that the OOD's reliability decreases as we gradually remove one of its components.

For MOON w/o LM, we set $w_f$ as 0.5, 0.001, and 0.00001, respectively, to incrementally remove the loss modification component. The AUROC (%) results for MOON w/o LM are summarized in the Table 13. $w_f = 1$ refers to our MOON.

Table 13: Additional incremental ablation study for loss modification (LM) by AUROC ($\uparrow$) in `MOON` with 95% sparse ResNet-18 for CIFAR-10.

| | $w_f = 1$ | $w_f = 0.5$ | $w_f = 0.001$ | $w_f = 0.00001$ |
|---|---|---|---|---|
| NearOOD | **88.71** | 86.97 | 86.52 | 85.76 |
| FarOOD | **91.62** | 90.18 | 89.41 | 87.76 |

For MOON w/o AT, we shorten $T_e$ to 3, 2, and 1, respectively, to incrementally remove the auto tuning component. The AUROC (%) results for MOON w/o AT are summarized in the Table 14. $T_e = 5$ refers to our MOON.

Table 14: Additional incremental ablation study for auto tuning (AT) by AUROC ($\uparrow$) in `MOON` with 95% sparse ResNet-18 for CIFAR-10.

| | $T_e = 5$ | $T_e = 3$ | $T_e = 2$ | $T_e = 1$ |
|---|---|---|---|---|
| NearOOD | **88.71** | 85.81 | 85.33 | 85.17 |
| FarOOD | **91.62** | 88.42 | 88.32 | 87.97 |

For MOON w/o VT, we shorten the length of epochs using the voting scheme at the end of training to 10, 5, and 2 epochs, respectively, to incrementally remove the voting scheme component. The AUROC (%) results for MOON w/o LM are summarized in the Table 15. Vote time = 20 refers to our MOON.

Table 15: Additional incremental ablation study for voting scheme (VT) by AUROC (↑) in MOON with 95% sparse ResNet-18 for CIFAR-10.

| | VOTE TIME = 20 | VOTE TIME = 10 | VOTE TIME = 5 | VOTE TIME = 2 |
|---|---|---|---|---|
| NearOOD | **88.71** | 85.84 | 85.32 | 84.77 |
| FarOOD | **91.62** | 88.16 | 86.78 | 86.76 |

# B  Appendix: Additional Details about Experiment Settings

## B.1  OOD Benchmark Description

The choice of out-of-distribution (OOD) Benchmarks is according to OpenOOD (Yang et al., 2022). The distribution here refers to the "label distribution". Following the common practice of constructing OOD detection benchmarks, we treat the entire dataset as in-distribution (ID) and then gather several datasets that are not correlated with any ID category as OOD datasets. We use four OOD benchmarks, which are named after ID datasets, including MNIST (Deng, 2012), CIFAR-10 (Krizhevsky et al., 2009), CIFAR-100 (Krizhevsky et al., 2009), and ImageNet (Russakovsky et al., 2015). As mentioned by OpenOOD (Yang et al., 2022), compared to the ID dataset, the near-OOD dataset has only semantic shifts, while the far OOD further contains significant covariate (domain) shifts. The detailed description of each benchmark is as follows.

**For MNIST** (Deng, 2012), it is a 10-class handwriting digit dataset with 60k images for training and 10k for test. Its near-OOD datasets include NOTMNIST (Yang et al., 2021b) and FashionMNIST (Hsu et al., 2020). The far-OOD datasets include Texture (Ahmed & Courville, 2020), CIFAR-10 (Tax, 2002) and TinyImageNet (Han et al., 2022), and Places-365 (Guo et al., 2017).

**For CIFAR-10** (Krizhevsky et al., 2009), it is a 10-class dataset for general object classification with 50k training images and 10k for test. Its near-OOD datasets include CIFAR-100 (Chandola et al., 2009) and TinyImageNet (Han et al., 2022), where 1,207 images are removed from TinyImageNet since they belong to CIFAR-10 classes (Yang et al., 2022). The far-OOD datasets include MNIST (Deng, 2012), SVHN (Netzer et al., 2011), Texture (Ahmed & Courville, 2020), and Places365 (Guo et al., 2017) with 1,305 images removed due to semantic overlaps.

**For CIFAR-100** (Krizhevsky et al., 2009), it is a 100-class dataset for general object classification with 50k training images and 10k for test. Its near-OOD datasets include CIFAR-10 (Tax, 2002) and TinyImageNet (Han et al., 2022), where 2,502 images are removed from TinyImageNet due to the overlapping semantics with CIFAR-100 classes (Yang et al., 2022). Its far-OOD include MNIST (Deng, 2012), SVHN (Netzer et al., 2011), Texture (Ahmed & Courville, 2020), and Places365 (Guo et al., 2017) with 1,305 images removed due to semantic overlaps.

**For ImageNet**, it is a large-scale image classification dataset with 1000 classes. OpenOOD (Yang et al., 2022) build the OOD dataset, where they use a 10k subset of Species (Torralba et al., 2008) with 713k images, iNaturalist (Shorten & Khoshgoftaar, 2019) with 10k images, ImageNet-O (Li et al., 2021) with 2k images, and OpenImage-O (Sun et al., 2022) with 17k images as near-OOD datasets, and use Texture (Ahmed & Courville, 2020), MNIST (Deng, 2012) as far-OOD. All images belonging to the ID classes are deleted.

## B.2  Computing Resources

The training and evaluation are mainly on NVIDIA Quadro RTX 6000. For CIFAR-10 and CIFAR-100, training ResNet-18 for 100 epochs with our MOON using 1 GPU takes around 1.5 hours. For ImageNet-12, training ResNet-50 for 100 epochs with our MOON using 1 GPU takes around 6 days.

## C  Appendix: Theoretical Analysis

In this section, we provide detailed proof of our insights in Section 4 of the main manuscript. The proof is based on Assumption C.1, which describes the characteristic of the feature space.

**Assumption C.1.** (Gaussian Mixture Feature Space): The feature mapping function $h$ maps the input ID data to a Gaussian mixture $v_1 \mathcal{N}(\mu_1, \Sigma_1) + v_2 \mathcal{N}(\mu_2, \Sigma_2)$. Specifically, when $y = 1$, we have $h(x) \sim \mathcal{N}(\mu_1, \Sigma_1)$. And when $y = 2$, we have $f(x) \sim \mathcal{N}(\mu_2, \Sigma_2)$.

### C.1  Insight 4.4: Unreliability

(Unreliability) suppose we have $(x_1, y_1)$ from class 1 and $D_2 = \{(x, y); ||h(x) - h(x_1)|| < \epsilon, y = 2\}$ from class 2. Then, unreliability can occur around $h(x_1)$.

*Proof*:

The final output probability is a composition of the feature mapping function $h$ and the softmax classification function $g$. As we know, the softmax function is Lipschitz continuous with a Lipschitz constant $c$ bounded by 1.

Because of the definition of $D_2$, we can know that

$$||g(h(x)) - g(h(x_1))|| \leq c||h(x) - h(x_1)||. \tag{4}$$

Suppose $g(h(x_1)) = \{p_{11}, p_{12}\}$ and $g(h(x_2)) = \{p_{21}, p_{22}\}$ for any $x_2 \in D_2$. Then we can know that

$$||p_{11} - p_{21}|| \leq c\epsilon. \tag{5}$$

Without loss of generality, suppose $x_1$ is predicted correctly, i.e., we have $p_{11} > 0.5$. Then, we can have

$$p_{21} \geq p_{11} - c\epsilon > 0.5, \ \epsilon \to 0. \tag{6}$$

Then, with small $\epsilon$, the samples in $D_2$ will have $p_{21} > 0.5$ and be incorrectly predicted to class 1. Thus, for any $\epsilon_0 < \epsilon$, we can define $D_2(\epsilon_0)$ and have its expected accuracy as:

$$D_2(\epsilon_0) = \{(x, y); ||h(x) - h(x_1)|| < \epsilon_0, y = 2\}, \tag{7}$$

$$\mathbf{E}_{D_2(\epsilon_0)}[1\{\hat{y} = y\}] = \mathbf{E}_{D_2(\epsilon_0)}[1\{1 = 2\}] = 0 \tag{8}$$

In this case, suppose the samples near $x_1$ are $D(\epsilon_0)$. And $D(\epsilon_0)$ can be divided into $D_1(\epsilon_0)$ and $D_2(\epsilon_0)$. $D_i = \{(x, y); ||h(x) - h(x_1)|| < \epsilon_0, y = i\}$. The ratio of the size of $D_1$ and $D_2$ can be approximated by

$$\frac{|D_1|}{|D_2|} = \frac{w_1}{w_2} = \frac{\mathcal{N}(h(x_1); \mu_1, \Sigma_1)}{\mathcal{N}(h(x_1); \mu_2, \Sigma_2)}, \text{ where } w_1 + w_2 = 1. \tag{9}$$

Then, we can calculate the unreliability level following the Definition 4.2:

$$\mathbf{E}_{D(\epsilon)}[\max_{c \in \{1,2\}} \mathcal{N}(h(x); \mu_c, \Sigma_c)] - \mathbf{E}_{D(\epsilon)}[1\{\hat{y} = y\}]$$

$$= w_1 \left( \mathbf{E}_{D_1(\epsilon)}[\max_{c \in \{1,2\}} \mathcal{N}(h(x); \mu_c, \Sigma_c)] - \mathbf{E}_{D_1(\epsilon)}[1\{\hat{y} = y\}] \right)$$

$$+ w_2 \left( \mathbf{E}_{D_2(\epsilon)}[\max_{c \in \{1,2\}} \mathcal{N}(h(x); \mu_c, \Sigma_c)] - \mathbf{E}_{D_2(\epsilon)}[1\{\hat{y} = y\}] \right)$$

$$= w_1 \left( \mathbf{E}_{D_1(\epsilon)}[\max_{c \in \{1,2\}} \mathcal{N}(h(x); \mu_c, \Sigma_c)] - 1 \right) + w_2 \left( \mathbf{E}_{D_2(\epsilon)}[\max_{c \in \{1,2\}} \mathcal{N}(h(x); \mu_c, \Sigma_c)] - 0 \right)$$

$$\geq p_{11} - c\epsilon - w_1$$

$$\tag{10}$$

Then, we can choose $x_1$ which is not very close to $\mu_1$ such that $w_1 < p_{11} - c\epsilon - \eta$. Then, we will have:

$$\mathbf{E}_{D(\epsilon)}[\max_{c \in \{1,2\}} \mathcal{N}(h(x); \mu_c, \Sigma_c)] - \mathbf{E}_{D(\epsilon)}[1\{\hat{y} = y\}] >\geq p_{11} - c\epsilon - w_1 > \eta. \tag{11}$$

The proof is complete.

### C.2 Insight 4.6: Hard-ID Reliability

(Hard-ID Reliability) Suppose we have the same $(x_1, y_1)$ and $D_2$ as in Insight 4.4. If the model is trained with our MOON method and the extra dimension successfully stores the unknown information, the unreliability can be solved, i.e.,

$$\mathbf{E}_{D(\epsilon)}[\max_{c \in \{1,2\}} \mathcal{N}(h(x); \mu_c, \Sigma_c)] - \mathbf{E}_{D(\epsilon)}[1\{\hat{y} = y\}] < \eta. \tag{12}$$

*Proof*:

Following the notations in the proof of Insight 4.4, we can know that unreliability around $x_1$ has two parts. One is from $D_1$ and the other is from $D_2$. And the majority of the unreliability comes from $D_2$. Thus, in order to reduce the discrepancy between confidence and accuracy, we need to mainly reduce the part of $D_2$, which is:

$$\mathbf{E}_{D_2(\epsilon)}[\max_{c \in \{1,2\}} \mathcal{N}(h(x); \mu_c, \Sigma_c)] - \mathbf{E}_{D_2(\epsilon)}[1\{\hat{y} = y\}] \tag{13}$$

Suppose the two classes of data are balanced. Since $x_1$ is predicted correctly, we can know that $w_1 > w_2$ and $|D_1| > |D_2|$.

In this case, if the model can find a better feature mapping $h$ that does not have such a discrepancy for any set of data $D$, then there is no safety problem. However, if finding such a suitable feature mapping $h$ is too difficult, which is usually the case, we can only change the softmax classification function $g$. In sparse training, the update route is cut off, generating spurious local optimization, leading to a more challenging exploration of the weight space and thus an inability to find a suitable feature mapping $h$.

If we choose the widely-used cross entropy loss $L$ which in our case will take the form as below:

$$
\begin{aligned}
L(D) &= L(D_1) + L(D_2) \\
&= -\sum_{i \in D_1} \log p_{i1} - \sum_{i \in D_2} \log p_{i2} \\
&= -\sum_{i \in D_1} \log p_{i1} - \sum_{i \in D_2} \log(1 - p_{i1})
\end{aligned}
\tag{14}
$$

Since it is difficult to find a better mapping $h$, we assume $h$ is fixed in this case. If $\epsilon$ is small, then the gradient of samples in $D(\epsilon)$ will be similar to that of $x_1$. Suppose $g(h(x_1)) = (p_1^*, p_2^*)$. Then, we can have:

$$L(D) = -\sum_{i \in D_1} \log p_{i1} - \sum_{i \in D_2} \log(1 - p_{i1}) \tag{15}$$

$$\approx -|D_1| \log p_1^* - |D_2| \log(1 - p_{i1}) \tag{16}$$

And the gradient of loss will be:

$$\nabla L(D) \propto \frac{(|D_2| + |D_1|) * p_1^* - |D_1|}{p_1^*(1 - p_1^*)} \tag{17}$$

Thus, to reduce the loss, $p_1^*$ will stuck at $\frac{|D_1|}{|D_1| + |D_2|}$ and can not reduce the discrepancy, implying severe unreliability in $D$.

If we choose our MOON , there will be three dimensions in the output probability and $g(h(x_1)) = (p_1^*, p_2^*, p_3^*)$. And the new loss will take the form as below:

$$
\begin{aligned}
L(D) &= L(D_1) + L(D_2) \\
&= -\sum_{i \in D_1} \log p_{i1} - \sum_{i \in D_2} \left(1 + \frac{w}{1 + w p_{i3}}\right) \log p_{i2} \\
&= -\sum_{i \in D_1} \log p_{i1} - \sum_{i \in D_2} \left(1 + \frac{w}{1 + w p_{i3}}\right) \log(1 - p_{i1}) \\
&\approx -|D_1| \log p_1^* - |D_2|(1 + \frac{w}{1 + w p_3^*}) \log(1 - p_1^*)
\end{aligned}
\tag{18}
$$

Then, the gradient of the loss will be as below

$$\nabla L(D) \propto \frac{(|D_1| + |D_2| + \frac{w}{1+wp_3^*}|D_2|) * p_1^* - |D_1|}{p_1^*(1 - p_1^*)} \tag{19}$$

We can see that in order to minimize the loss, we can update not only $p_1^*$ but also $p_3^*$, thus providing more degrees of freedom for the model to learn reliable predictions. Instead of increasing $p_1^*$ and being stuck in an unreliable state, the model can increase $p_3^*$ when it encounters hard-ID samples. Suppose we have nonzero $p_3^*$ after the parameter update. We can calculate the new unreliability level following the Definition 4.2:

$$\mathbf{E}_{D(\epsilon)}[\max_{c\in\{1,2\}} \mathcal{N}(h(x); \mu_c, \Sigma_c)] - \mathbf{E}_{D(\epsilon)}[1\{\hat{y} = y\}]$$

$$= w_1 \left( \mathbf{E}_{D_1(\epsilon)}[\max_{c\in\{1,2\}} \mathcal{N}(h(x); \mu_c, \Sigma_c)] - \mathbf{E}_{D_1(\epsilon)}[1\{\hat{y} = y\}] \right)$$

$$\quad + w_2 \left( \mathbf{E}_{D_2(\epsilon)}[\max_{c\in\{1,2\}} \mathcal{N}(h(x); \mu_c, \Sigma_c)] - \mathbf{E}_{D_2(\epsilon)}[1\{\hat{y} = y\}] \right)$$

$$= w_1 \left( \mathbf{E}_{D_1(\epsilon)}[\max_{c\in\{1,2\}} \mathcal{N}(h(x); \mu_c, \Sigma_c)] - 1 \right) + w_2 \left( \mathbf{E}_{D_2(\epsilon)}[\max_{c\in\{1,2\}} \mathcal{N}(h(x); \mu_c, \Sigma_c)] - 0 \right)$$

$$\leq w_1(p_{11} + c\epsilon - p_3^* - 1) + w_2(p_{11} + c\epsilon - p_3^*)$$

$$= p_{11} + c\epsilon - p_3^* - w_1 \tag{20}$$

Since the extra dimension (i.e., $p_3^*$) successfully stores the unknown information, we can assume that $p_3^*$ is reasonably large, i.e., $p_3^* > p_{11} + c\epsilon - w_1 - \eta$. Then, the unreliability level can be reduced to a value less than $\eta$:

$$\mathbf{E}_{D(\epsilon)}[\max_{c\in\{1,2\}} \mathcal{N}(h(x); \mu_c, \Sigma_c)] - \mathbf{E}_{D(\epsilon)}[1\{\hat{y} = y\}] \leq p_{11} + c\epsilon - p_3^* - w_1 < \eta \tag{21}$$

The proof is complete.

### C.3 Insight 4.8: OOD Reliability

(OOD Reliability) Suppose we achieve Hard-ID Reliability in Insight 4.8 with our `MOON` method, we can have lower confidence on OOD data, implying stronger OOD detection.

*Proof*:

We know that hard-to-detect OOD data $D_3$ usually have features close to those of hard ID data $D_2$, which is similar to the scenario described in Insight 4.4. Otherwise, if the features of the hard-to-detect OOD data are very different from those of hard ID data, then the OOD data becomes easy to detect without the OOD unreliability.

Thus, these hard-to-detect ID samples can be viewed as pseudo-OOD data. And the $\{h(x_2); x_2 \in D_2\}$ are close to $\{h(x_3); x_3 \in D_3\}$. Then we can follow the proof of Insignt 4.6 and show that:

$$\mathbf{E}_{D_3(\epsilon)}[\max_{c\in\{1,2\}} \mathcal{N}(h(x); \mu_c, \Sigma_c)] \text{ is reduced.} \tag{22}$$

In this way, we are able to have smaller confidence in the hard-to-detect OOD data $D_3$, which helps to enhance the OOD reliability. This describes how we can extract unknown information from hard ID data, explaining the outlier-exposure-free property of `MOON`.

## D   Appendix: Limitations of Our `MOON`

### D.1   More General Reliability

Despite the enhanced OOD reliability achieved in sparse training, `MOON` does not address this matter comprehensively from a broader reliability perspective, including robust generalization (Silva & Najafirad,

2020; Özdenizci & Legenstein, 2021; Chen et al., 2022a) and adaptation (Emam et al., 2021; Alayrac et al., 2022; Hu et al., 2022), which are equally vital aspects of reliability and deserve further investigation (Tran et al., 2022).

### D.2 Reliability in Large Generative Models

Additionally, the effectiveness of MOON remains unexplored in the context of widely employed large generative models (Ouyang et al., 2022; Croitoru et al., 2023). On the one hand, as model size expands, novel changes in model characteristics may arise (Wei et al., 2022b). On the other hand, given the prevalence of non-classification tasks within LLMs, the exploration of how to evaluate their reliability remains an uncharted domain.

## E Broader Impact

This work is likely to encourage other future work on more general reliability in sparse training to achieve efficiency and comprehensive reliability in real-world DNN deployments.

In addition, as large generative models develop, both computational efficiency and decision reliability are important aspects of their application. This work can increase the progress of efficient and reliable large generative models.

## F FLOPs Analysis

In this section, we add FLOPs analysis to MOON. We use 80% sparse ResNet-50 on Image-Net-2012 as an example. The training FLOPs using RigL are about 7.4e17 (Evci et al., 2020). For our MOON, the increased FLOPs come from three components, i.e., loss modification, auto tuning, and voting scheme.

(i) For loss modification and auto-tuning.

- In the early stage where $w = 0$, we have about 1.3e5 FLOPs in each iteration because of the computation of $\beta$.

- After the early stage, from the calculation of $w$ and the new loss, we have about 2.6e5 FLOPs in each iteration.

- Thus, the loss modification and auto tuning introduce an additional 2.3e15 FLOPs.

(ii) For voting scheme,

We start doing weight averaging in 80% of the training epoch and have about 1.8e9 FLOPs from the weight averaging. In each epoch after the 80% training epoch, we also have about 2.4e15 FLOPs from the batchNorm buffer updates. Thus, the voting scheme introduces an additional 4.8e16 FLOPs.

Overall, our MOON introduces an additional 5.0e16 FLOPs, which is about 6.8% of the original RigL's FLOPs. The main cause of the increase in FLOPs is the update of the batchNorm buffer. To further reduce the increase of FLOPs, we can reduce the update frequency of the batchNorm buffer.

We also conduct experiments to compare the training time using our MOON and baseline methods. We train 80%, 90%, 95%, and 99% sparse ResNet-18 on both CIFAR-10 and CIFAR-100 using MOON + RigL and RigL. We denote the training time of our benchmark method, RigL, as 1 to help understand exactly what overhead our MOON adds to the training process. The results are summarized in the Table 16-17. We can see that MOON adds less than 5% overhead.

Table 16: Training time comparison using our `MOON` + RigL and RigL with 80%, 90%, 95%, and 99% sparse ResNet-18 on both CIFAR-10.

|  | CIFAR-10 (80%) | CIFAR-10 (90%) | CIFAR-10 (95%) | CIFAR-10 (99%) |
|---|---|---|---|---|
| RigL | 1 | 1 | 1 | 1 |
| MOON + RigL | 1.045 | 1.042 | 1.039 | 1.046 |

Table 17: Training time comparison using our `MOON` + RigL and RigL with 80%, 90%, 95%, and 99% sparse ResNet-18 on both CIFAR-100.

|  | CIFAR-10 (80%) | CIFAR-10 (90%) | CIFAR-10 (95%) | CIFAR-10 (99%) |
|---|---|---|---|---|
| RigL | 1 | 1 | 1 | 1 |
| MOON + RigL | 1.049 | 1.046 | 1.048 | 1.043 |

## G   Hyperparameter Tuning

The hyperparameters of the baseline method follow their original settings in (Evci et al., 2020; Yang et al., 2022). The hyperparameters of our `MOON` are described as follows.

(i) For the learning rate, following (Yang et al., 2022), we use the cosine annealing scheduler, where the maximum learning rate is set to 1.2 and the minimum learning rate is set to 0.08.

(ii) For the fraction of weights removed and added (pruning ratio) in each sparse pattern update, we follow the original settings in (Evci et al., 2020; Sundar & Dwaraknath, 2021).

- The initial fraction is 0.3.
- This fraction decays in the mask update step by cosine annealing.
- The score is set to zero after 70% of the training cycles.

(iii) For the weight distribution, we follow the original settings in (Evci et al., 2020; Sundar & Dwaraknath, 2021) and initialize the mask by Erdos-Renyi-Kernel (ERK).

(iv) For $w_f$ and $r$

- For MNIST, we set $w_f$ and $r$ as 0.01 and 64, respectively.
- For CIFAR-10, we set $w_f$ and $r$ as 1 and 64, respectively.
- For CIFAR-100, we set $w_f$ and $r$ as 2 and 64, respectively.
- For ImageNet-2012, we set $w_f$ and $r$ as 0.001 and 64, respectively.

## H   Feature Visualization

One of the assumptions on which our theory is based is that the feature distribution of a DNN feature extractor is a Gaussian mixture. Additional experiments are added to support this assumption. Specifically, we train ResNet-18 on CIFAR-10 with 80%, 90%, 95% and 99% sparsity and collect features before the last layer. Then, to visualize the feature distribution, we use the PCA method to reduce the dimensionality and plot the histogram of the first component.

As shown in Figure 9-12, the feature distribution shows a high density in the center and then decreases towards the two sides across all classes and sparsity levels. This characterization is consistent with a Gaussian distribution. Therefore, the overall distribution for all classes can be approximated by a Gaussian mixture distribution, supporting our assumption.

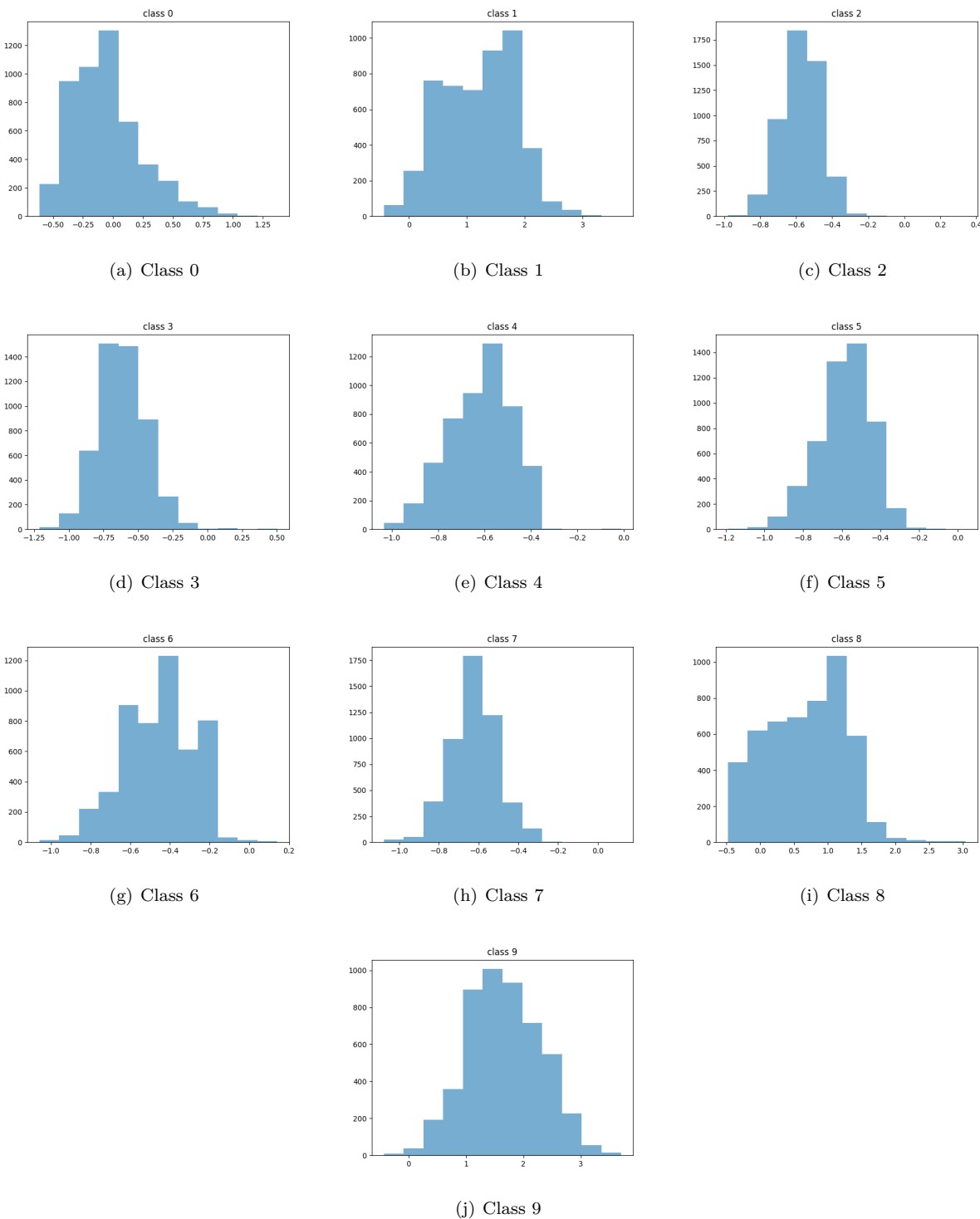

Figure 9: Feature distribution visualization on CIFAR-10 for 80% sparse ResNet-18. The feature distribution shows a high density in the center and then decreases towards the two sides across all classes.

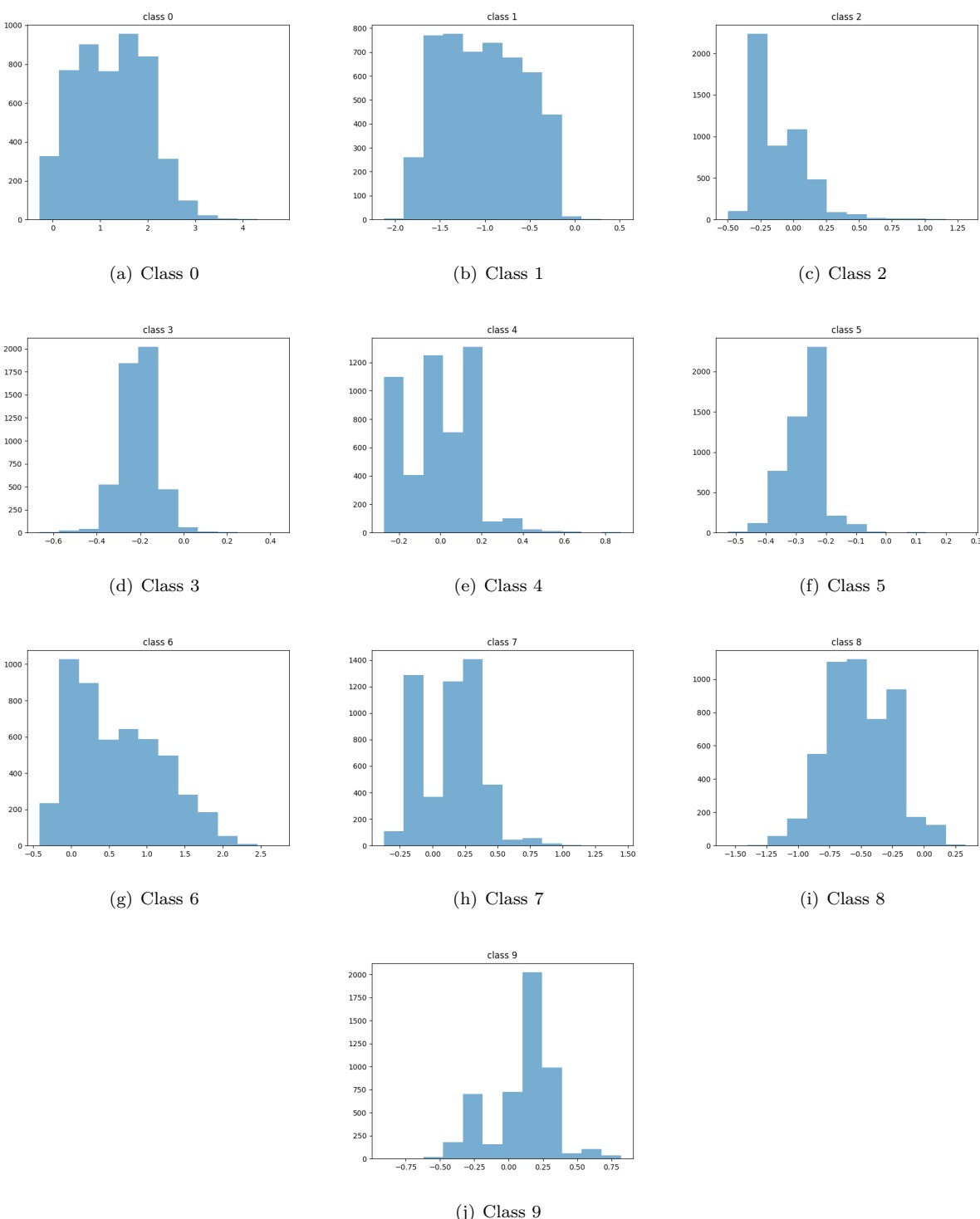

Figure 10: Feature distribution visualization on CIFAR-10 for 90% sparse ResNet-18. The feature distribution shows a high density in the center and then decreases towards the two sides across all classes.

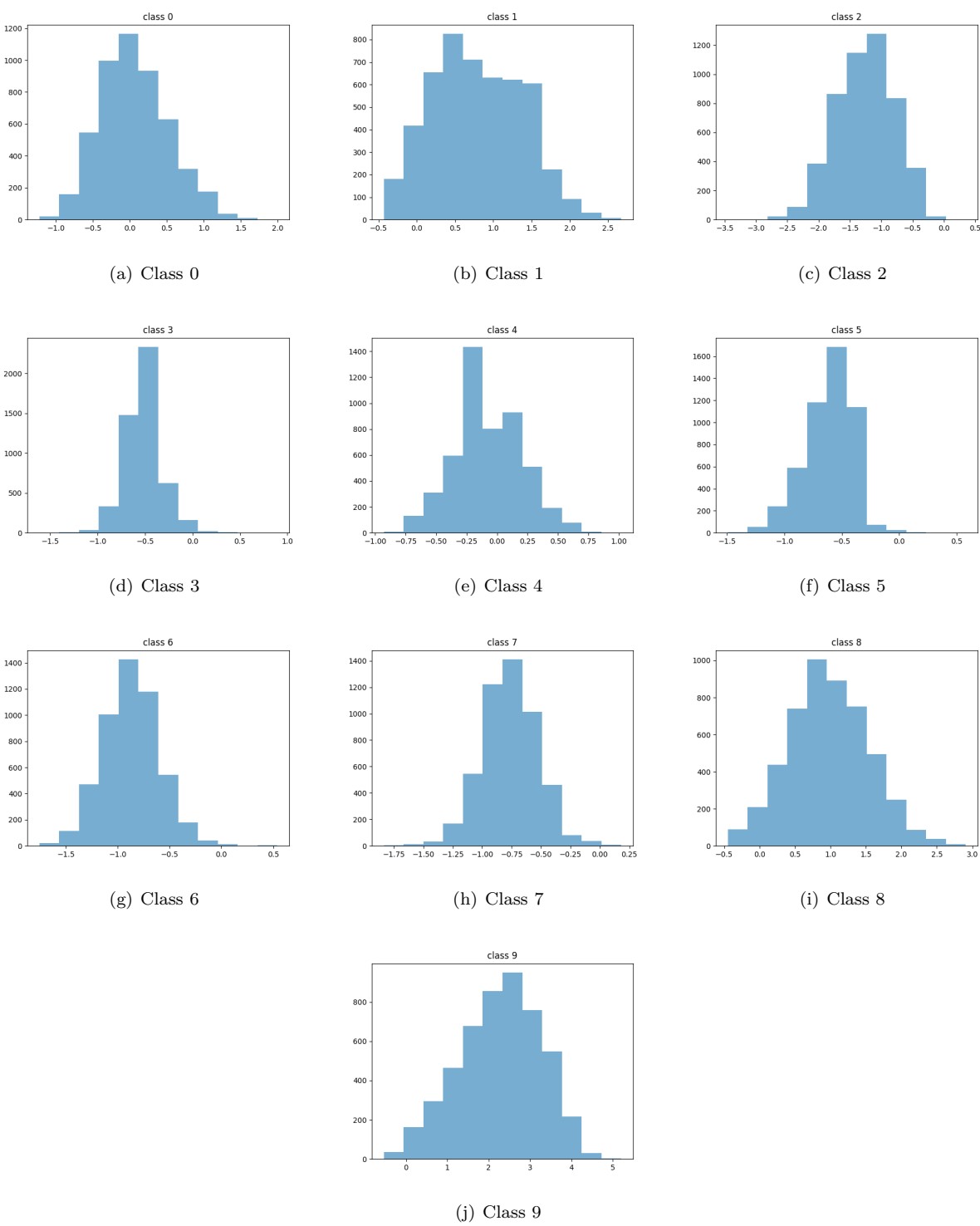

Figure 11: Feature distribution visualization on CIFAR-10 for 95% sparse ResNet-18. The feature distribution shows a high density in the center and then decreases towards the two sides across all classes.

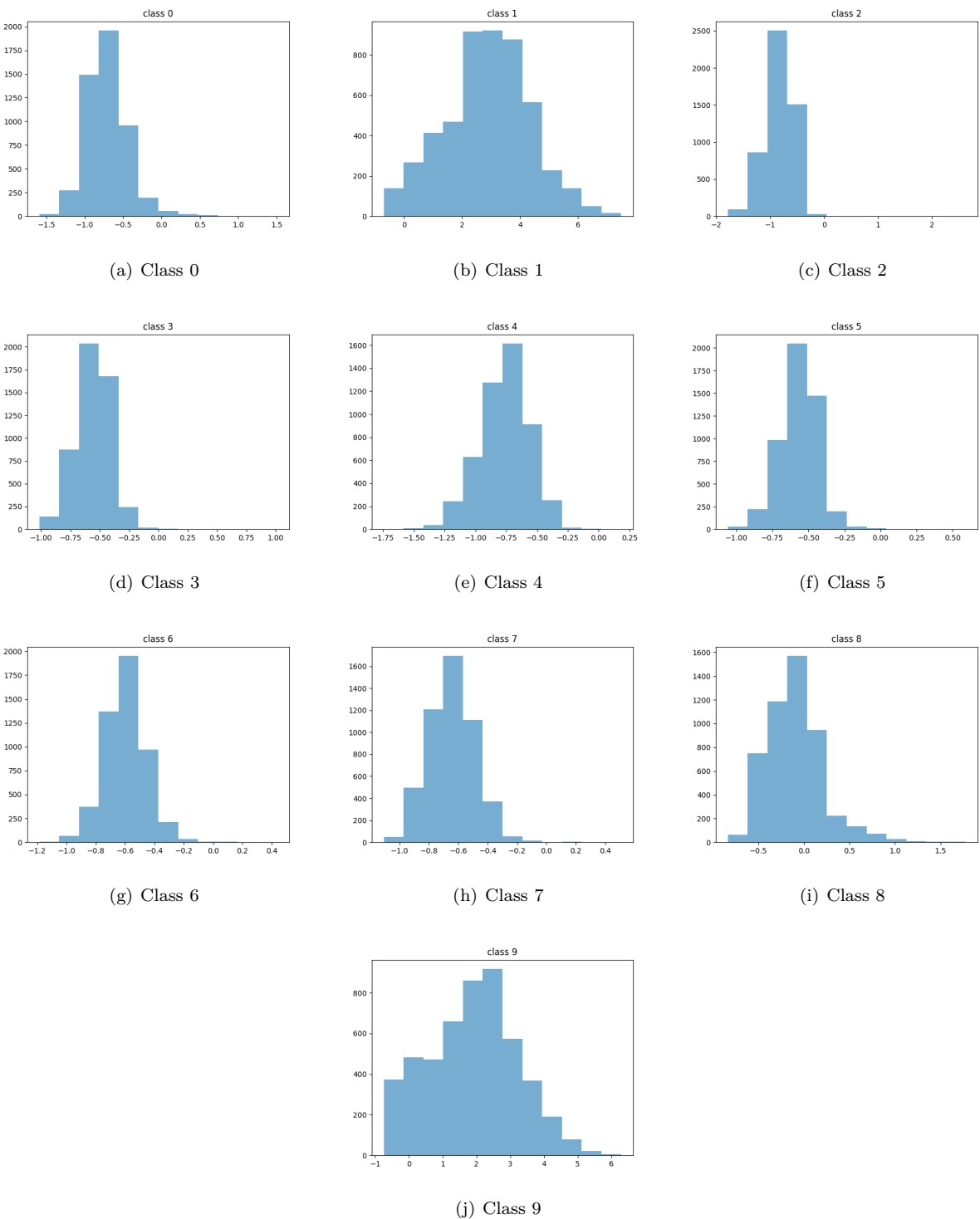

Figure 12: Feature distribution visualization on CIFAR-10 for 99% sparse ResNet-18. The feature distribution shows a high density in the center and then decreases towards the two sides across all classes.

