# OpenReview forum: "Embracing Unknown Step by Step: Towards Reliable Sparse Training in Real World"
_TMLR — Accepted by TMLR_

### Review · Reviewer_yuD9 · 2023-11-19

**Summary Of Contributions:**

The submission presents a new sparse training method called MOON, which aims to improve the reliability of deep neural networks (DNNs) in detecting out-of-distribution (OOD) data. It identifies the problem and proposes to leverage unknown information under sparse constraints to effectively guide weight space exploration to an OOD-reliable state. Experiments show that MOON can effectively enhance the reliability of OOD detection in DNNs.

**Audience:**

Yes

**Claims And Evidence:**

Yes

**Requested Changes:**

see the weaknesses part above.

**Strengths And Weaknesses:**

Strengths:

- The proposed MOON method effectively leverages unknown information under sparse constraints to improve real-world reliability in DNNs

- The K+1-way formulation allows the model to distinguish between known and unknown information, and the unknown-aware loss modification provides an effective guide for weight space exploration during sparse training.

- The theoretical insights provided in the paper offer a deeper understanding of the reliability issues and the effectiveness of the MOON method in addressing them.

- The experimental results demonstrate the improved OOD detection capabilities of the MOON method.

Weaknesses:

- The explanation of key algorithm 1 requires further clarification. Specifically, it is unclear whether the auto-tuning of $w$ occurs after each epoch or during each sample's training. If it's the former, the meaning and role of the probabilities $p_i, 1\le i \le K+1$ need clarification. Additionally, the initial factor $r$ is mentioned but not updated or explained in the context of the algorithm. The expression $L(p_1,\cdots,p_{K+1})$ is also confusing, as $L$ is initially described as operating on $x_i$ in equation (1).

- The paper repeatedly claims that the MOON method introduces minimal cost compared to other methods, but this assertion lacks empirical support. It would be beneficial to include experiments or data showing the specific overheads MOON adds to the training process and how these compare to other existing methods.

- The OOD setting discussed seems limited to classification problems, with the unknown-aware loss constrained to cross-entropy loss. It would be valuable to explore whether there's a more general framework capable of incorporating the high-level concepts of this method, applicable to a broader range of contexts.

---

> ### Author Response · Authors · 2024-01-12
> **Response to reviewer yuD9: Q1 - Q3**
>
> Thank you for your valuable review. Please see our responses below to clarify the main concerns.
>
> **Q1: The explanation of key algorithm 1 requires further clarification. Specifically, it is unclear whether the auto-tuning of $w$ occurs after each epoch or during each sample's training. If it's the former, the meaning and role of the probabilities $p_i$ need clarification. Additionally, the initial factor $r$ is mentioned but not updated or explained in the context of the algorithm. The expression $L(p_1,\cdots,p_{K+1})$ is also confusing, as $L$ is initially described as operating on $x_i$ in equation (1).**
>
> A1: We have updated the manuscript to make it clearer. The clarifications are as follows.
>
> (i) Algorithm 1 is the auto-tuning pipeline of $w$, and $w$ is used to control the amount of unknown information we incorporate into the model. The detailed explanation of the auto-tuning is as below.
> * $w$ is set as 0 in the first 5 epochs.
> * Then, we set $w=w_i$ at the beginning of the 6-th epoch.
> * During each epoch, $w$ will be kept as a constant.
> * When an epoch ends, we increase $w$ by $\delta$ so that it gradually increases to $w_f$.
>
> (ii) The role of the probability term (i.e., $p_{y_i}$ and $p_{K+1}$) is to help the model learn and store both the known and unknown information.
> * During the training, the model aims to reduce the loss which takes the form of $-(1+\frac{w}{1+wp_{K+1}})\log p_{y_i}$.
> * To minimize loss, the model can increase $p_{y_i}$ so that the model makes correct predictions. This means that the model learns the known information.
> * When the sample is a hard ID sample, it might not be possible to increase $p_{y_i}$. In this case, the model can increase $p_{K+1}$ to reduce the loss, which indicates that the model learns the unknown information.
>
> (iii) For the initial factor $r$, it is used to control the initial value $w_i$ to avoid $w$ being too large. We set it to 64 in our experiments. This strategy is inspired by [R1].
>
> (iv) For the expression $L(p_1,\cdots,p_{K+1})$, the reason that we write it in this way is $p_1,\cdots,p_{K+1}$ is a function of $x_i$, i.e., $p_1,\cdots,p_{K+1}=f(x_i)$.
> * We have change $L(p_1,\cdots,p_{K+1})$ to $L(x_i)$ avoid confusion.
>
> [R1] Combating Label Noise in Deep Learning Using Abstention, ICML 2019.
>
> **Q2: The paper repeatedly claims that the MOON method introduces minimal cost compared to other methods, but this assertion lacks empirical support. It would be beneficial to include experiments or data showing the specific overheads MOON adds to the training process and how these compare to other existing methods.**
>
> A2: We have added additional experiments to compare the training time using our MOON and baseline methods.
> * We train 80%, 90%, 95%, and 99% sparse ResNet-18 on both CIFAR-10 and CIFAR-100 using MOON + RigL and RigL.
>
> * We denote the training time of our benchmark method, RigL, as 1 to help understand exactly what overhead our MOON adds to the training process.
> * The results are summarized in the table below. We can see that MOON adds less than 5% overhead.
>
> |  | CIFAR-10 (80%) | CIFAR-10 (90%) | CIFAR-10 (95%) | CIFAR-10 (99%) |
> | :---------: | :-----: | :-----: | :-----: | :-----: |
> | RigL | 1 | 1 | 1 | 1 |
> | MOON + RigL | 1.045 | 1.042 | 1.039 | 1.046 |
>
> |  | CIFAR-100 (80%) | CIFAR-100 (90%) | CIFAR-100 (95%) | CIFAR-100 (99%) |
> | :---------: | :-----: | :-----: | :-----: | :-----: |
> | RigL | 1 | 1 | 1 | 1 |
> | MOON + RigL | 1.049 | 1.046 | 1.048 | 1.043 |
>
> **Q3: The OOD setting discussed seems limited to classification problems, with the unknown-aware loss constrained to cross-entropy loss. It would be valuable to explore whether there's a more general framework capable of incorporating the high-level concepts of this method, applicable to a broader range of contexts.**
>
> A3: Our work focuses on OOD reliability in classification tasks.
> * Regression does not have the softmax structure and is a completely different problem
> * OOD reliability in regression tasks is a good future research direction.
>
> Intuitively, from a high-level perspective, to improve OOD reliability in regression tasks, we can continue working on the feature space.
> * We can use the hard ID data as the pseudo-hard OOD data to provide the model with unknown information.
> * However, since we do not have the softmax structure in regression tasks, how to include the unknown information is not trivial and needs future research.

---

### Review · Reviewer_KgVE · 2023-12-20

**Summary Of Contributions:**

The paper proposed a new training loss for multi-class classification called MOdel-agnostic unknOwN-aware sparse training (MOON). Training with MOON will boost the network's ability in detecting out-of-distribution (OOD) samples. MOON augmented the label space to include an extra label that means "unknown". This turns the K-ary classification problem to K+1-ary classification problem. To ensure the model can assign proper probabilities to the "unknown" class, MOON adjusts the cross-entropy loss by assigning a larger weights $(1 + w_i / (1 + w_{K+1}))$ to hard samples that model gave incorrect predictions. In addition, the author proposed to gradually adjust the weights during training to give the model enough time to learn the easy samples. MOON can be combined with other sparse training and OOD detection methods. Experiments show that MOON consistently boosts the OOD detection performance.

**Audience:**

Yes

**Claims And Evidence:**

No

**Requested Changes:**

1. The author needs to mention related work in calibrated learning and also compare with those methods.
2. Clarify the connection between the theoretical insights and the MOON algorithm.
3. Clarify the potential errors in the proof.
4. Conduct ablation study on the w_f.

**Strengths And Weaknesses:**

Strengths:

1. The idea of adding an extra label that means "unknown" and reweighing the samples based on p_{unknown} is intuitive.
2. Performance of MOON is good according to the experiments.


Weaknesses:

1. K+1-ary softmax has been adopted in calibrated learning, such as "[ICML2020] Consistent Estimators for Learning to Defer to an Expert" and "[ICML2022] Calibrated Learning to Defer with One-vs-All Classifiers". The author has not mentioned these relevant works.
2. It is not clear how the theoretical insights in Section 4 associates with the MOON algorithm. For example, I cannot see why the author picked (1 + w_i / (1 + w_{K+1})) as the weights by reading Section 4.
3. I cannot understand the proof of the Insight 4.4 (Appendix C.1) and Insight 4.6 (Appendix C.2). Specifically, in Appendix C.1, I cannot understand why equation (6) is correct. I can see that $p_{21} \geq p_{11} - c \epsilon$ . Since $p_{11} > 0.5$ , this gives $p_{21} \geq p_{11} - c \epsilon \gt 0.5  - c \epsilon$ . I cannot see why $p_{21}$ is larger than 0.5. The equation (14) in Appendix C.2 does not seem to be correct. $\log p_{i2}$ should be equal to $\log (1 - p_{i1})$ and does not equal to $1 - \log p_{i1}$.
4. The sensitivity of w_f is unclear. The author reported the hyper-parameter choice of w_f in Appendix G but there is no ablation study.

---

> ### Author Response · Authors · 2024-01-12
> **Response to reviewer KgVE: Q1 - Q3**
>
> Thank you for your valuable review. Please see our responses below to clarify the main concerns.
>
> **Q1: K+1-ary softmax has been adopted in calibrated learning, such as [R5] and [R6]. The author has not mentioned these relevant works.**
>
> A1: We have discussed K+1-ary softmax in Section 2 about related work on "Extra Dimension", and we add [R5, R6] in this related work section.
>
> They are all very different from our method.
> * Existing K+1-ary softmax methods do not directly exploit the unknown information and incur extra costs, or require exposure to OOD data.
> * In addition, they all ignore how to add extra dimensions in sparse training.
> * As for [R5], it requires access to additional information from experts and does not consider sparse training.
> * As for [R6], it mainly focuses on ID calibration and also does not take into account sparse training.
>
> [R5] Consistent Estimators for Learning to Defer to an Expert, ICML2020.
>
> [R6] Calibrated Learning to Defer with One-vs-All Classifiers, ICML2022.
>
> **Q2: It is not clear how the theoretical insights in Section 4 associates with the MOON algorithm. For example, I cannot see why the author picked (1 + w_i / (1 + w_{K+1})) as the weights by reading Section 4.**
>
> A2: The theoretical insights tell us how our MOON uses unknown information to guide the model training to improve the OOD's reliability.
>
> (i) Definition 4.2 gives a mathematical expression for the ID unreliability in feature space around $h(x_0)$.
>
> (ii) Insight 4.4 justifies the existing unreliability in a two-way classification problem around $h(x_1)$.
>
> (iii) Insight 4.6 proves that our MOON can solve the unreliability in Insight 4.4 and achieve hard-ID reliability.
> * Detailed proof is included in Appendix C.2.
> * Intuitively, when using close-world softmax loss function, for hard-ID samples, it is difficult for the model to further increase the probability for the true class.
> * Thus, the model is unable to further reduce the loss and will be stuck at unreliable predictions for the hard-ID samples.
> * In contrast to the softmax function, by using the unknown information and with $(1 + w_i / (1 + w_{K+1}))$, our MOON can enable the model to further reduce the loss via increasing $w_{K+1}$, instead of the probability for the true class.
> * This can allow the model to provide more reliable predictions for the hard-ID samples.
> * This unknown information from the hard ID samples can give the model some insight into the hard OOD samples, thus improving the OOD reliability.
>
> **Q3: I cannot understand the proof of the Insight 4.4 (Appendix C.1) and Insight 4.6 (Appendix C.2). Specifically, in Appendix C.1, I cannot understand why equation (6) is correct. I can see that $p_{21} \geq p_{11} - c\epsilon$. Since $p_{11} > 0.5$, this gives $p_{21} \geq p_{11} - c\epsilon$. I cannot see why $p_{21}$ is larger than 0.5. The equation (14) in Appendix C.2 does not seem to be correct. $\log p_{i2}$ should be equal to $\log (1 - p_{i1})$ and does not equal to $1 - \log p_{i1}$.**
>
> A3: (i) For equation (6) in Appendix C.1, we have $p_{21} \geq p_{11} - c\epsilon$ and $p_{11} > 0.5$. Also, we have assumed that $\epsilon \rightarrow 0$. Suppose $e=p_{11} - 0.5$. As long as $\epsilon$ is small enough, we can have $\epsilon < \frac{e}{c}$. Thus, we have $c\epsilon < e = p_{11} - 0.5$. This leads to $p_{11} - c\epsilon > 0.5$. Then, we can get $p_{21} \geq p_{11} - c\epsilon > 0.5$, i.e., $p_{21} \geq 0.5$.
>
> (ii) For equation (14) in Appendix C.2, we have corrected it to $\log p_{i2}=\log (1 - p_{i1})$ at Appendix C.2 in the manuscript. The proof pipeline is similar.
> * We first discuss the unreliability in $D_2$.
> * Then, we examine the softmax loss function and its gradient and show that the softmax loss function can put the model into an unreliable state.
> * Finally, we investigate the situation using our MOON. We find that our MOON is able to provide more degrees of freedom for the model to learn reliable predictions.
> * Reducing the loss by adding probability values for additional dimensions can reduce the level of unreliability.

---

> > ### Author Response · Authors · 2024-01-12
> > **Response to reviewer KgVE: Q4**
> >
> > **Q4: The sensitivity of $w_f$ is unclear. The author reported the hyper-parameter choice of $w_f$ in Appendix G but there is no ablation study.**
> >
> > A4: We add experiments on CIFAR-10 to test the sensitivity of $w_f$.
> > * As summarized in Appendix G, we set $w_f=1$ for CIFAR-10.
> > * We add experiments of 95% sparse ResNet-18 with $w_f=0.1, 0.5, 2, 4$.
> > * The AUROC results are summarized in the table below. We also include them in Appendix A.4.
> > * As shown in the table, when we increase $w_f$ from 0.1 to 4, OOD reliability first increases and then decreases after 1.
> > * While OOD reliability may be degraded when $w_f$ is too small or too large, OOD reliability is not degraded much when $w_f$ is between 0.5 and 2.
> > * The proper $w_f$ value can be found with a simple hyperparameter tuning.
> >
> > | | CIFAR-100 | TIN | NearOOD | MNIST | SVHN | Texture | Places365 | FarOOD |
> > | :---------: | :-----: | :-----: | :-----: | :-----: | :-----: | :-----: | :-----: | :-----: |
> > |   $w_f=0.1$   | 83.57 | 85.25 | 84.41 | 86.49 | 86.47 | 85.35 | 84.73 | 85.76 |
> > |   $w_f=0.5$   | 86.25 | 87.07 | 87.66 | 93.46 | 91.02 | 87.80 | 85.11 | 89.35 |
> > |   $w_f=1$    | 88.04 | 89.39 | 88.71 | 94.70 | 93.35 | 89.86 | 88.51 | 91.62 |
> > |   $w_f=2$   | 86.45 | 87.77 | 87.11 | 89.96 | 91.44 | 89.24 | 86.05 | 89.17 |
> > |   $w_f=4$    | 84.78 | 85.84 | 85.31 | 94.69 | 91.44 | 86.66 | 84.65 | 89.36 |

---

### Review · Reviewer_Ww7e · 2023-12-28

**Summary Of Contributions:**

The authors explore the phenomenon of reliability of OOD detection in DNN models trained with sparsity constraints.The authors hypothesize that sparse DNN models might suffer from OOD unreliability due to their constrained weight space exploration (as many of the weights go to zero leading to limited exploration in the weight space). The authors then present a uncertainty aware loss formulation for training sparse DNNs by introducing an extra dimension in the softmax classification layer of DNNs which is representative of the uncertainty in the output prediction. Based on the training stage (early or late), an auto-tuning strategy for tuning the weight for this loss is also presented. Finally, the authors combine multiple models (using weight averaging) from different training epochs to get a final prediction.

**Audience:**

Yes

**Claims And Evidence:**

Yes

**Requested Changes:**

Questions:

1. While the theoretical insights are intuitive, as a non-expert in this field, I wanted to know about the practical usefulness of Assumption 4.1. To assume the distribution of features from a DNN feature extractor to be a gaussian mixture with the same number of classes seems a bit simplified. It would be great if the authors can comment on this. It seems like this assumption can be validated empirically by clustering the last layer features for a pretrained DNN?

2. For weight averaging, what strategy does the method use? Is it a traditional uniform averaging or weighted averaging of some kind (for instance, with more preference for models at later epochs.) This should be clarified in Section 3.3.

3. From the ablation experiments in Section 5.7, it looks like the method suffers most without voting. Is there any intuition behind this? Also I would encourage the authors to present an incremental ablation where each component of the method is added incrementally over the previous component which could provide more intuition on the utility of each component.

Minor weaknesses

1. “However, it has been shown that DNNs tend to focus on the known due to the close nature of the training process and ignore the rest”-Missing citations here.

2. Typo: Under “Unknown-aware Loss Modification: Specifically, in the new loss, we add a larger weight”, there is a mismatch in the expression for the weight from Eq 1 and the expression for the weight in the inline equation in this line.

**Strengths And Weaknesses:**

Strengths:

1. Exhaustive empirical results on datasets with varying complexity.
2. The method is post-hoc and modular and therefore easy to integrate with existing strategies for sparse training (as is also illustrated in the experiments)
3. I found the presentation clear in the sense that the assumptions for the theoretical intuitions are clearly stated with proofs in the Appendix. The experimental settings are also clearly described.

---

> ### Author Response · Authors · 2024-01-12
> **Response to reviewer Ww7e: Q1 - Q3 (i)**
>
> Thank you for the insightful comments. Please see our responses below to clarify the main concerns.
>
> **Q1: While the theoretical insights are intuitive, as a non-expert in this field, I wanted to know about the practical usefulness of Assumption 4.1. To assume the distribution of features from a DNN feature extractor to be a gaussian mixture with the same number of classes seems a bit simplified. It would be great if the authors can comment on this. It seems like this assumption can be validated empirically by clustering the last layer features for a pretrained DNN?**
>
> A1：(i) The assumption that the feature distribution of DNN feature extractors is a Gaussian mixture has been widely used in OOD detection studies [R1, R2, R3].
> * This simplification is a good approximation of more complex real-world data.
> * This approximation can support more theoretical studies and help us gain a more in-depth understanding of the OOD reliability in DNNs.
>
> (ii) Theoretical studies of more complex distributions in OOD work are premature.
> * Using more complex distributions requires new theoretical tools that are beyond the scope of this paper, which can be a good direction for future work.
>
> (iii) We add additional experiments to validate and support this assumption.
> * We train ResNet-18 on CIFAR-10 with 80%, 90%, 95% and 99% sparsity.
> * We collect features before the last layer.
> * To visualize the feature distribution, we use the PCA method to reduce the dimensionality and plot the histogram of the first component.
> * As shown in Figures 9-12 in Appendix H, the feature distribution shows a high density in the center and then decreases towards the two sides across all classes and sparsity levels.
> * This characterization is consistent with a Gaussian distribution. Therefore, the overall distribution for all classes can be approximated by a Gaussian mixture distribution.
>
> [R1] A simple unified framework for detecting out-of-distribution samples and adversarial attacks, NeurIPS 2018.
>
> [R2] Probabilistic modeling of deep features for out-of-distribution and adversarial detection, 2019.
>
> [R3] Provable guarantees for understanding out-of-distribution detection, AAAI 2022.
>
> **Q2: For weight averaging, what strategy does the method use? Is it a traditional uniform averaging or weighted averaging of some kind (for instance, with more preference for models at later epochs.) This should be clarified in Section 3.3.**
>
> A2: The averaging scheme is a uniform averaging among the model weights near the end of training. We have updated the Section 3.3 to make it more clear.
>
> (i) More specifically, the averaging scheme includes the following steps:
> * First, in the first 80% of training epochs, we do not use weight averaging.
> * Second, during the later 20% epochs, we collect models at each epoch.
> * Then, we take a uniform averaging among the collected models to get the final output single sparse model with improved OOD reliability.
>
> (ii) The uniform averaging scheme is inspired by the weighted averaging method, which is widely used in existing studies and shows good performance [R4, R5].
>
> [R4] Model soups: averaging weights of multiple fine-tuned models improves accuracy without increasing inference time, ICML 2022.
>
> [R5] Calibrating the Rigged Lottery: Making All Tickets Reliable, ICLR 2023.
>
> **Q3: From the ablation experiments in Section 5.7, it looks like the method suffers most without voting. Is there any intuition behind this? Also, I would encourage the authors to present an incremental ablation where each component of the method is added incrementally over the previous component which could provide more intuition on the utility of each component.**
>
> A3: (i) Intuitively, the reason the method suffers the most without voting in some scenarios is that voting combines information to better identify what is known and what is unknown.
> * Loss modification and auto-tuning strategy gradually tell the model the unknown information.
> * However, only part of the unknown information is available for each model collected, resulting in suboptimal information.
> * Highly sparse DNNs are sensitive to misleading information, and incomplete unknown information can be harmful, leading to degraded performance and reliability.
> * The voting scheme can then combine different known and unknown information from the collected models to give the final output a more comprehensive understanding of the known and unknown information.

---

> ### Author Response · Authors · 2024-01-12
> **Response to reviewer Ww7e: Q3 (ii) - Q3 (v)**
>
> A3: (ii) Voting schemes, loss modification, and auto-tuning strategy are all important and indispensable to improve the reliability of sparse training.  Existing work supports that this voting scheme does not significantly improve reliability.
> * Our voting scheme is based on the weight-averaging method.
> * Existing work [R1] mentions that while weight averaging produces better generalization, it does not improve confidence calibration.
> * This suggests that the reliability improvement comes not only from the weight averaging method (voting scheme), but also from loss modification and automatic adjustment.
>
> (iii) Our ablation study in Section 5.7 shows the importance of each component in our MOON. The results for ID and OOD are shown in Table 7 and Figure 5, respectively.
> * For loss modifications, the MOON without loss modifications shows an increase in ECE and FPR-95.
> * For auto-tuning, the MOON without auto-tuning shows an increase in ECE and FPR-95.
> * For the voting scheme, the MOON without the voting scheme shows an increase in ECE and FPR-95 and a decrease in accuracy.
>
> (iv) We add an incremental ablation study in which each component of the method is incremented to the previous component to better intuitively understand the utility of each component.
>
> * We train 95% sparse ResNet-18 on CIFAR-10 with MOON + RigL. We find that the OOD's reliability decreases as we gradually remove one of its components.
> * For MOON w/o LM, we set $w_f$ as 0.5, 0.001, and 0.00001, respectively, to incrementally remove the loss modification component. The AUROC (%) results for MOON w/o LM are summarized in the table below. $w_f$ = 1 refers to our MOON.
>
> |  | $w_f$ = 1 | $w_f$ = 0.5 | $w_f$ = 0.001 | $w_f$ = 0.00001 |
> | :---------: | :-----: | :-----: | :-----: | :-----: |
> | NearOOD | **88.71** | 87.66 | 86.52 | 85.76 |
> | FarOOD | **91.62** | 89.35 | 89.27 | 87.76 |
>
> * For MOON w/o AT, we shorten $T_e$ to 3, 2, and 1, respectively, to incrementally remove the auto tuning component. The AUROC (%) results for MOON w/o AT are summarized in the table below. $T_e$ = 5 refers to our MOON.
>
> |  | $T_e$ = 5 | $T_e$ = 3 | $T_e$ = 2 | $T_e$ = 1 |
> | :---------: | :-----: | :-----: | :-----: | :-----: |
> | NearOOD | **88.71** | 85.81 | 85.33 | 85.17 |
> | FarOOD | **91.62** | 88.42 | 88.32 | 87.97 |
>
> * For MOON w/o VT, we shorten the length of epochs using the voting scheme at the end of training to 10, 5, and 2 epochs, respectively, to incrementally remove the voting scheme component. The AUROC (%) results for MOON w/o LM are summarized in the table below. Vote time = 20 refers to our MOON.
>
> |  | vote time = 20 | vote time = 10 | vote time = 5 | vote time = 2 |
> | :---------: | :-----: | :-----: | :-----: | :-----: |
> | NearOOD | **88.71** | 85.84 | 85.32 | 84.77 |
> | FarOOD | **91.62** | 88.16 | 86.78 | 86.76 |
>
> (v) We further add experiments to compare our MOON with RigL with the voting scheme on CIFAR-10 and CIFAR-100. Our MOON has better reliability on both ID and OOD data and maintains accuracy compared to RigL with voting scheme.
> * For ID calibration, as shown in the table below, our MOON shows smaller ECE values (smaller is better) compared to RigL with the voting scheme, which implies a more calibrated prediction for ID data.
>
> |  |   CIFAR-10: 90% |   CIFAR-10: 95%   |  CIFAR-10: 99%  |  CIFAR-100: 90% |   CIFAR-100: 95%   |   CIFAR-100: 99%   |
> | :---------: | :-----: | :-----: | :-----: | :-----: | :-----: | :-----: |
> |    RigL+Vote    | 0.0291 | 0.0292 | 0.0185 | 0.1011 | 0.0811 | 0.0536 |
> |   MOON   | **0.0207** | **0.0173** | **0.0094** | **0.0849** | **0.0666** | **0.0132** |
>
> * For OOD detection, our MOON has a smaller FPR-95 and larger AUROC compared to RigL with voting scheme, which means more effective OOD detection for OOD data. The FPR-95 results for CIFAR-10 are shown in the table below (smaller is better).
>
> | | CIFAR-100 | TIN | NearOOD | MNIST | SVHN | Texture | Places365 | FarOOD |
> | :---------: | :-----: | :-----: | :-----: | :-----: | :-----: | :-----: | :-----: | :-----: |
> |   RigL+Vote (90%)   | 63.79 | 60.18 | 61.99 | 43.63 | **50.14** | 60.00 | 59.52 | 53.32 |
> |   MOON (90%)   | **62.70** | **58.18** | **60.44** | **33.67** | 54.65 | **57.32** | **58.41** | **51.01** |
> |   RigL+Vote (95%)   | 65.53 | 61.64 | 63.59 | 42.41 | 59.10 | **61.58** | 62.95 | 56.51 |
> |   MOON (95%)   | **65.29** | **61.11** | **63.20** | **39.26** | **52.72** | 63.44 | **62.38** | **54.45** |
> |   RigL+Vote (99%)   | 72.18 | 70.18 | 71.18 | 53.78 | 74.61 | 67.77 | 70.46 | 66.65 |
> |   MOON (99%)   | **70.38** | **67.05** | **68.72** | **44.34** | **72.08** | **62.75** | **68.54** | **61.93** |

---

> > ### Author Response · Authors · 2024-01-12
> > **Response to reviewer Ww7e: Q3 (v) - Q5**
> >
> > A3: (v) AUROC results on CIFAR-10 are shown in the Table below (larger is better).
> >
> > | | CIFAR-100 | TIN | NearOOD | MNIST | SVHN | Texture | Places365 | FarOOD |
> > | :---------: | :-----: | :-----: | :-----: | :-----: | :-----: | :-----: | :-----: | :-----: |
> > |   RigL+Vote (90%)   | 88.73 | 89.77 | 89.25 | 94.16 | 93.47 | 90.39 | 89.78 | 91.95 |
> > |   MOON (90%)   | **88.92** | **90.34** | **89.63** | **95.39** | **93.09** | **91.09** | **89.92** | **92.37** |
> > |   RigL+Vote (95%)   | 88.07 | 89.39 | 88.73 | 94.30 | 92.23 | 89.68 | 88.77 | 91.24 |
> > |   MOON (95%)   | **88.04** | **89.39** | **88.71** | **94.76** | **93.35** | **89.86** | **88.51** | **91.62** |
> > |   RigL+Vote (99%)   | 83.50 | 84.46 | 83.98 | 89.96 | 87.46 | 84.77 | 83.52 | 86.43 |
> > |   MOON (99%)   | **83.75** | **85.11** | **84.43** | **91.96** | **89.01** | **86.44** | **83.38** | **87.70** |
> >
> > * For ID performance, our MOON has similar accuracy compared to RigL with voting scheme, implying comparable ID performance.
> >
> > |  |   CIFAR-10: 90% |   CIFAR-10: 95%   |  CIFAR-10: 99%  |  CIFAR-100: 90% |   CIFAR-100: 95%   |   CIFAR-100: 99%   |
> > | :---------: | :-----: | :-----: | :-----: | :-----: | :-----: | :-----: |
> > |    RigL+Vote    | 94.21 | 93.47 | 89.49 | 75.13 | **73.31** | 62.77 |
> > |   MOON   | **94.43** | **93.70** | **89.66** | **75.18** | 73.10 | **63.17** |
> >
> > [R1] Wortsman, Mitchell, et al. "Model soups: averaging weights of multiple fine-tuned models improves accuracy without increasing inference time." ICML, 2022.
> >
> > **Q4: “However, it has been shown that DNNs tend to focus on the known due to the close nature of the training process and ignore the rest”-Missing citations here.**
> >
> > A4: This statement can be referred to [R2, R3, R4]. We have added it to the manuscript.
> >
> > [R2] Wang, Yezhen, et al. "Energy-based open-world uncertainty modeling for confidence calibration." ICCV, 2021.
> >
> > [R3] Abhijit Bendale and Terrance E Boult. Towards open set deep networks. In CVPR, 2016.
> >
> > [R4] Shreyas Padhy, Zachary Nado, Jie Ren, Jeremiah Liu, Jasper Snoek, and Balaji Lakshminarayanan. Revisiting one-vs-all classifiers for predictive uncertainty and out-ofdistribution detection in neural networks. arXiv preprint arXiv:2007.05134, 2020
> >
> > **Q5: Typo: Under “Unknown-aware Loss Modification: Specifically, in the new loss, we add a larger weight”, there is a mismatch in the expression for the weight from Eq 1 and the expression for the weight in the inline equation in this line.**
> >
> > A5: We have corrected the typo.

---

> > > ### Comment · Reviewer_Ww7e · 2024-01-19
> > > **Reviewer Response**
> > >
> > > Thanks to the authors for presenting additional results and intuitions which clarify most of my concerns.

---

### Author Response · Authors · 2024-01-12
**Common response to all reviewers**

We thank all reviewers for their insightful comments. The main contributions of this work as stated by the reviewers are as follows:

* The proposed outlier-exposure-free method is intuitive [Reviewer KgVE]
* The method is post-hoc and easy to integrate with existing sparse training strategies [Reviewer Ww7e, KgVE]
* The method provides an effective way to utilize unknown information for model training [Reviewer KgVE, yuD9]
* The method is well-explained with theoretical insight [Reviewer Ww7e, yuD9]
* Solid experiments show clear improvements over existing methods [Reviewer Ww7e, KgVE, yuD9]
* The paper is well written, clearly motivated, and of high quality [Reviewer Ww7e]

We would like to emphasize the significant contribution of this paper to sparse training. While being applied in a wide variety of applications, sparse training’s OOD reliability remains a crucial concern. This work provides the first investigation of how sparse training influences OOD reliability and analyzes the hidden reasons, paving the way toward applying sparse training in real-world decision-making systems. In the paper, we reveal the issue of weaker OOD detection in existing sparse training methods, and propose a simple yet effective algorithm to greatly improve the OOD detection while still keeping the high accuracy. We have provided both theoretical insights and extensive experimental results over different datasets and architectures to demonstrate the effectiveness of our method. Our method only incurs a small overhead and can be used as a drop-in replacement for existing sparse training methods. We believe this work fills an important gap in sparse training and hope the reviewers can consider our response in their final evaluation.

---

### Decision · Action_Editor_PZfM · 2024-03-02

**Recommendation:** Accept as is

**Comment:**

This paper investigates the reliability of sparse training from the out-of-distribution (OOD) perspective. The paper reveals that sparse training exacerbates OOD unreliability and proposes a training method to enhance sparse training for detecting OOD data. The training method includes the modifying of the loss function by introducing an extra dimension in the softmax layer, incorporating an auto-tuning strategy and a voting (model averaging) scheme. The effectiveness of the methods is empirically evaluated across a variety of dataset, model architectures and sparsity levels, alongside theoretical insights.

The paper received recommendations of one accept, one leaning accept and one leaning reject. Overall, reviewers acknowledge several strengths of the paper:

1). Exploring a new area, i.e. reliability of OOD detection with sparse training.

2). The proposed methods are sound and intuitive.

3). The empirical evaluation is extensive, demonstrating consistent improvement in OOD detection reliability across various benchmarks.

Reviewers raised two main concerns:

1). The provided theory seemed overly simplified and somewhat confusing.

2). The novelty of the paper is somewhat limited as it primarily used existing methods for sparse training.

During rebuttal, the authors provided clarification on the theory and additional data to justify the assumptions. The authors also conducted additional experiments, including ablation experiments and an assessment of training overhead, to further justify their claims and understanding of the methods.

In summary, while the novelty is somewhat limited and the proposed method at this stage can be only used for classification problems, this paper is technically solid and potential benefits in the domain of sparse training. Therefore, I recommend acceptance.

**Audience:**

Yes. The insight and techniques provided in the paper will be beneficial in the area of sparse training.

**Claims And Evidence:**

Yes. The claims are supported by clear evidence, especially after the authors provided additional clarification and experiment results during rebuttal.